# RSCC: A Large-Scale Remote Sensing Change Caption Dataset for Disaster Events

**Zhenyuan Chen**[1][*]   **Chenxi Wang**[2]   **Ningyu Zhang**[2]   **Feng Zhang**[1,3,4][†]

bili_sakura@zju.edu.cn   sunnywcx@zju.edu.cn
zhangningyu@zju.edu.cn   zfcarnation@zju.edu.cn

[1]School of Earth Sciences, Zhejiang University, Hangzhou 310058, China
[2]School of Software Technology, Zhejiang University
[3]Zhejiang Provincial Key Laboratory of Geographic Information Science, Hangzhou 310058, China
[4]Key Laboratory of Spatio-temporal Information and Intelligent Services (LSIIS), Ministry of
Natural Resources of the People's Republic of China

## Abstract

Remote sensing is critical for disaster monitoring, yet existing datasets lack temporal image pairs and detailed textual annotations. While single-snapshot imagery dominates current resources, it fails to capture dynamic disaster impacts over time. To address this gap, we introduce the Remote Sensing Change Caption (RSCC) dataset, a large-scale benchmark comprising 62,351 pre-/post-disaster image pairs (spanning earthquakes, floods, wildfires, and more) paired with rich, human-like change captions. By bridging the temporal and semantic divide in remote sensing data, RSCC enables robust training and evaluation of vision-language models for disaster-aware bi-temporal understanding. Our results highlight RSCC's ability to facilitate detailed disaster-related analysis, paving the way for more accurate, interpretable, and scalable vision-language applications in remote sensing. Code and dataset are available at `https://github.com/Bili-Sakura/RSCC`.

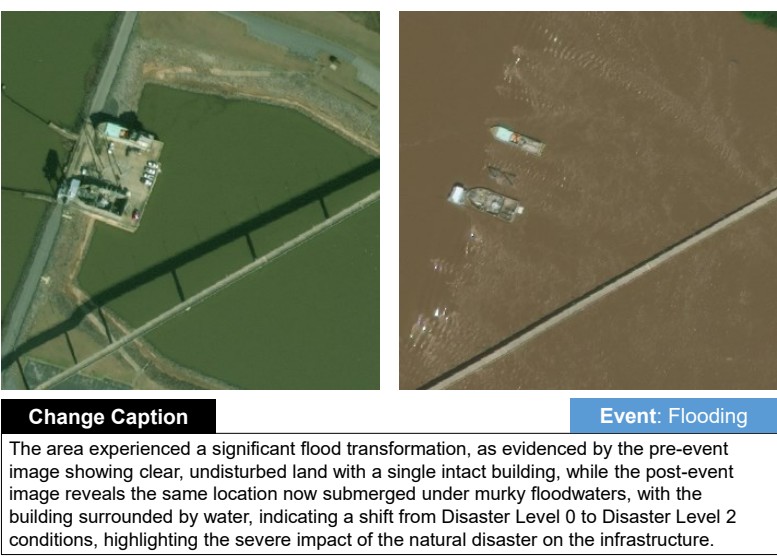

**Change Caption**                                    **Event**: Flooding

The area experienced a significant flood transformation, as evidenced by the pre-event image showing clear, undisturbed land with a single intact building, while the post-event image reveals the same location now submerged under murky floodwaters, with the building surrounded by water, indicating a shift from Disaster Level 0 to Disaster Level 2 conditions, highlighting the severe impact of the natural disaster on the infrastructure.

Figure 1: A sample from RSCC dataset.

---

[*]First author

[†]Corresponding author

39th Conference on Neural Information Processing Systems (NeurIPS 2025) Track on Datasets and Benchmarks.

# 1 Introduction

Temporal remote sensing imagery is indispensable for monitoring dynamic Earth processes, particularly disaster events that demand rapid response and analysis. Temporal remote sensing data has proven indispensable in supporting disaster relief planning and response [62, 61, 21]. However, the inherently complex spatiotemporal relationships embedded within this data pose significant challenges for effective analysis and interpretation.

Advancements in the modeling of multimodal data have enabled generalist Multimodal Large Language Models (MLLMs) [1, 2, 52, 51, 20, 19, 42, 56, 75, 45] that can perform a variety of natural image interpretation tasks specified flexibly through natural language. Specifically, MLLMs trained in a interleaved way have a deep visual-semantic understanding across images [11, 32, 47, 74, 79, 33]. These models achieve great success in multi-image reasoning [44, 72, 82] and video understanding [41, 8, 7, 23, 16], while their capabilities in temporal remote sensing image understanding remain underexplored.

Existing remote sensing image-text datasets often focus on single-snapshot imagery and lack the temporal details vital for understanding dynamic events, particularly in disaster-related scenarios, As shown in Table 1. Although there are multi-temporal datasets (e.g ., fMoW [12], SpaceNet 7 [69], S2Looking [64], QFabric [70] and SpaceNet 8 [22]), none of them provide rich textual descriptions of how scenes change over time. However, their potential for disaster-specific temporal analysis remains untapped due to the absence of high-quality bi-temporal datasets with detailed textual annotations. Existing remote sensing datasets either focus on generic land-use changes or provide short captions lacking disaster context. For instance, LEVIR-CC [37] annotates urban development but omits disaster-specific details, while Dubai-CCD [25] offers brief descriptions without capturing nuanced damage levels or infrastructure transformations.

Table 1: Comparison with existing remote sensing text-image datasets.

| Dataset | Year | #Image (Pixels) | Caption #Captions (Avg_L) | Caption Details | Temporal |
|---|---|---|---|---|---|
| UCM-Captions [54] | 2016 | 2,100 (1.0B) | 10,500 (12) | ✗ | ✗ |
| RSICD [43] | 2018 | 10,921 (0.5B) | 54,605 (12) | ✗ | ✗ |
| fMoW [12] | 2018 | 1M (437.0B) | N/A | ✗ | ✓ |
| SpaceNet 7 [69] | 2021 | 2,389 (2.6B) | N/A | ✗ | ✓ |
| S2Looking [64] | 2021 | 5,000 (5.0B) | N/A | ✗ | ✓ |
| QFabric [70] | 2021 | 2,520 (245.1B) | N/A | ✗ | ✓ |
| SpaceNet 8 [22] | 2022 | 2,576 (3.0B) | N/A | ✗ | ✓ |
| LEVIR-CC [37] | 2022 | 20,154 (1.2B) | 50,385 (40) | ✓ | ✓ |
| Dubai-CCD [25] | 2022 | 1,000 (<0.1B) | 2,500 (35) | ✓ | ✓ |
| RSICap [26] | 2023 | 2,585 (0.6B) | 2,585 (60) | ✓ | ✗ |
| RS5M [81] | 2024 | 5M (-) | 5M (49) | ✓ | ✗ |
| VRSBench [34] | 2024 | 29,614 (7.8B) | 29,614 (52) | ✓ | ✗ |
| WHU-CDC [65] | 2024 | 14,868 (1.9B) | 37,170 (-) | ✓ | ✓ |
| XLRS-Bench [73] | 2025 | 934 (67.5B) | 934 (379) | ✓ | ✗ |
| **RSCC (Ours)** | 2025 | 124,702 (32.7B) | 62,351 (72) | ✓ | ✓ |

To address these challenges, we introduce the Remote Sensing Change Caption (RSCC) dataset, the first large-scale dataset tailored for disaster-aware bi-temporal understanding . RSCC bridges critical gaps by:

1. Large-Scale Event-Driven Dataset : 62,351 pre-/post-disaster image pairs sourced from 31 global events, spanning earthquakes, floods, wildfires, and more.

2. A specialized model for remote sensing change captioning: To validate the robustness of our dataset, we train a MLLM specialized for remote sensing change captioning based on

RSCC dataset. The benchmark result shows that RSCC dataset enhance the capabilities of general MLLMs on remote sensing temporal image understanding.

3. Change Caption Benchmark : We develop a change caption benchmark based on our RSCC dataset and evaluate the performance of several state-of-the-art temporal MLLMs.

The remainder of this paper is organized as follows. In Section 2, we detail the construction process of RSCC, including data sources and caption generation pipeline. Section 3 introduce the our specialzed remote sensing change captioning model trained on RSCC dataset. In Section 4, we benchmark existing temporal MLLMs' change captioning capabilities on RSCC and presents both qualitative and quantitative results.

## 2   Pipeline

To construct our RSCC dataset, we employed a multimodal reasoning model - Qwen QvQ-Max [59] - along with existing human label to generate high fidelity captions. QvQ-Max is the latest proprietary MLLM that is capable of visual reasoning which shows superior capabilities in zero-shot remote sensing image change caption (see Appendix A). Unlike traditional MLLMs that prioritize recognition-based outputs, QvQ-Max leverages a structured reasoning process to infer spatial-temporal relationships [5]. The QvQ-Max captioning process takes about \$5/k image pairs. The overall dataset construction pipeline is shown in Figure 2.

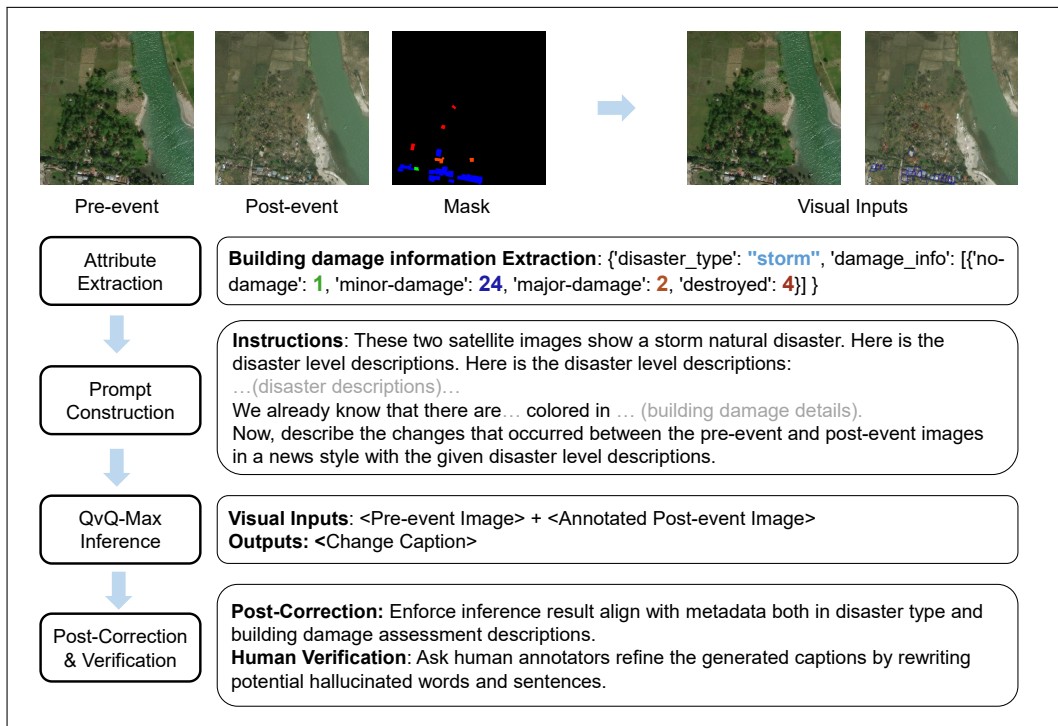

Figure 2: Illustration of RSCC dataset construction pipeline. We extract building damage information from labels and use carefully designed instructions to prompt QvQ-Max with reasoning capabilities and generate change captions from input images with building damage information.

### 2.1   Data Source

In this study, we utilize xBD dataset [21] along with EBD dataset [77], which are all obtained from MAXAR OpenData Program. The images are cropped without overlapping to 512×512 from xBD's original 1024×1024, while EBD retains its 512×512 resolution. The overall RSCC datasets consists of 62,351 bi-temporal pre- and post-disaster image pairs (xBD: 44,136; EBD: 18,215) spanning from 31 events covering disaster types ranging from earthquake, flooding (hurricane), tsunami, storm (hurricane, tornado), volcano eruption and wildfire. Full events list is shown in Appendix A.

## 2.2 Attribute Extraction

The xBD dataset contains human annotations of building bounding boxes with damage assessment labels. The damage evaluation is based on the Joint Damage Scale [21], which was developed with contributions from organizations such as NASA and the California Air National Guard. This scale is designed to assess building damage from satellite imagery across various disaster scenarios, providing detailed descriptions for different level ranging from no damage to destroyed.

## 2.3 Prompt Construction

We carefully design the following instructions to prompt QvQ-Max [59] to create detailed bi-temporal image change captions. We convert building damage labels into in-context auxiliary information. Shtedritski *et al.* [66] found that by applying marking-based visual prompt engineering, it is possible to unlock effective behaviors in vision-language models like CLIP [60], even without any training examples. This approach led to state-of-the-art results in zero-shot referring expression comprehension tasks. Inspired by this idea, we construct building damage masks as visual prompts for MLLMs.

The prompt for QvQ-Max consists of visual inputs and textual inputs (instructions) (Figure 2). The visual inputs are composed of original pre-event image and annotated post-event image where building bounding boxes are added onto the post-event image with different color that denote the damage level . The textual inputs are formatted as <task instructions> <disaster descriptions> <building damage details> and <output format> . The complete visual prompt template is shown in Appendix A.

## 2.4 QvQ-Max Inference

Given input prompts, we call QvQ-Max (qvq-max-2025-03-25) API from Alibaba Cloud [3] to automatically generate annotations. For xBD dataset change caption generation, we fix the prompt as one discussed in Section 2.3 which yield the optimal results in the empirical study. As EBD dataset does not contain human labeled annotation, we use naive prompt as "<pre_image><post_image>You will be provided with two satellite images of the same area before and after a {disaster_type} natural disaster event. Describe the changes in a news style with a few sentences". We do not observe any issue of instruction mis-following or invalid output format for captions generated from both datasets.

## 2.5 Post-Correction and Human Verification

To ensure the reliability of captions generated by QvQ-Max, we implement a two-stage post-correction process. First, the Qwen2.5-Max [57] systematically enforces metadata alignment by correcting discrepancies in disaster type (e.g., resolving mismatches between "hurricane" and metadata-specified "flooding") and damage descriptions (e.g., revising "minor damage" to "destroyed" based on building annotations). This automated stage achieves disaster type consistency with metadata from 93.2% to 100.0%. Second, a subset of RSCC captions (10%) is randomly selected and manually validated by three experts using a 0/1 binary rubric across four criteria: disaster type accuracy, damage detail completeness, factual consistency, and clarity. 100.0% of sampled captions passed validation. Failed captions were reprocessed through the automated pipeline with refined rules, ensuring final dataset consistency. Full details of correction rules and evaluation protocols are provided in Appendix A.

# 3 RSCC Dataset

## 3.1 Overview

Our RSCC dataset comprises a total of 62,315 bi-temporal image pairs, each annotated with a detailed change caption. These image pairs capture a range of real-world disaster scenarios, reflecting a diverse set of geographical locations, disaster types, and severity levels. By offering rich textual descriptions of scene changes, RSCC aims to facilitate advanced temporal reasoning and caption generation tasks for large vision-language models. A summary of these caption statistics is detailed in Figure 3.

---

[3] https://bailian.console.aliyun.com/

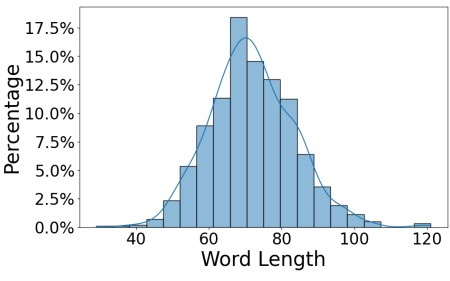
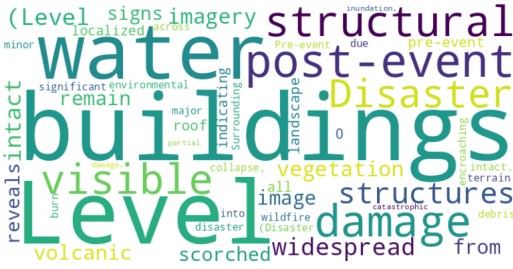

| (a) Number of words distribution. | (b) Wordcloud of captions. |

Figure 3: Statistics of RSCC.

## 3.2 RSCC for vision-language model training

In order to facilitate the vision-language model training, we divide the RSCC dataset into two splits, where a *train* split contains 61,363 image pairs from 31 distinct events across xBD as well as EBD and a test split contains 988 image pairs from 19 distinct events in xBD. We conduct full-parameter fine-tuning on Qwen2.5-VL 7B [58] using *train* set for 2 epoch with a $batch\_size = 1$ on a single node equipped with 2 NVIDIA H800 GPUs. We initialize the learning rate at 1e-6 and 1e-5 for LLM backbone and vision encoder respectively and employ a cosine learning rate decay schedule for optimization. For image inputs, we maintain the native resolution of RSCC as 512×512 for maximum pixel inputs and minimum pixel inputs. The training procedure cost a total of 40 GPU clockwall hours.

## 4 Benchmark Evaluation

### 4.1 Experiment Settings

**Baselines.** For remote sensing change captioning, we benchmark moderate size open-sourced MLLMs (less than 13B parameters) that supports multi-image inputs, including LLaVA-NeXT-Interleave [33], xGen-MM[4] (BLIP-3) [79], LLaVA-OneVision [32], Qwen2-VL [74],Pixtral [47] Phi-4-Multimodal [46] Kimi-VL [29] and InternVL 3 [84]. We also add two specialized remote sensing change captioning models namely TEOChat [27] and CCExpert [76].

**Evaluation Metrics.** For model evaluation, we compare the text similarity with n-gram overlap metrics including ROUGE [36] and METEOR [4]. While the aforementioned measures are commonly reported in image captioning works, we find they are suboptimal to measure the semantic similarity across long texts. Therefore, we follow Kaggle LLM Prompt Recovery Competition [5] and introduce Sentence T5-XXL Embedding [49] with Sharpened Cosine Similarity [6] (ST5-SCS) to get a well-established similarity measure. We set $q = 0$ and $p = 3$ for sharpened cosine similarity.

**MLLMs Configurations.** For model generation, we use the default sampling strategy derived from configuration. We use the same prompt style in section 2 for xBD dataset and omit building damage assessment information for EBD dataset. We compare the performance of change captions by three settings (i.e. zero-shot, textual prompt and visual prompt). We evaluate on RSCC *test* set. More implement details are shown in Appendix A.

### 4.2 Quantitative Results

Our evaluation on the RSCC dataset reveals three primary insights into image captioning performance (Table 2):

1. **Model Scale vs. Performance** The performance of vision-language models for remote sensing change captioning generally improves with increased parameter count, as seen in LLaVA-NeXT-

---

[4] https://hugggingfae.co/Salesforce/xgen-mm-phi3-mini-instruct-interleave-r-v1.5
[5] https://www.kaggle.com/c/llm-prompt-recovery

Interleave (8B) achieving 46.99% ST5-SCS and Qwen2-VL (7B) reaching 45.55% ST5-SCS . However, Kimi-VL (3B) exceeds expectations with 51.35% ST5-SCS , indicating that architectural optimizations or domain-specific tuning can mitigate limitations in model size. Larger proprietary models like InternVL 3 (8B) and Pixtral (12B) dominate metrics such as ROUGE (19.87% ) and ST5-SCS (79.18% ), though open-source models remain competitive baselines.

Table 2: Detailed image caption performance on the subset of RSCC dataset (naive/zero-shot results). Avg_L denotes the average word number of generated captions. **Boldface** indicates the best performance while underline denotes the suboptimal performance. *BLIP-3 and LLaVA-OneVision tend to repeat their answer endlessly, which cause large caption lengths.

| Model | N-Gram | | Contextual Similarity | Avg_L |
|---|---|---|---|---|
| (#activate params) | ROUGE(%)↑ | METEOR(%)↑ | ST5-SCS(%)↑ | |
| BLIP-3 (3B) [79] | 4.53 | 10.85 | 44.05 | *456 |
| Kimi-VL (3B)[29] | 12.47 | 16.95 | 51.35 | 87 |
| Phi-4-Multimodal (4B) [46] | 4.09 | 1.45 | 34.55 | 7 |
| Qwen2-VL (7B)[74] | 11.02 | 9.95 | 45.55 | 42 |
| LLaVA-NeXT-Interleave (8B) [33] | 12.51 | 13.29 | 46.99 | 57 |
| LLaVA-OneVision (8B)[32] | 8.40 | 10.97 | 46.15 | *221 |
| InternVL 3 (8B) [84] | 12.76 | 15.77 | 51.84 | 64 |
| Pixtral (12B) [47] | 12.34 | 15.94 | 49.36 | 70 |
| CCExpert (7B) [76] | 7.61 | 4.32 | 40.81 | 12 |
| TEOChat (7B)[27] | 7.86 | 5.77 | 52.64 | 15 |
| **Ours (7B)** | **14.99** | **16.05** | **58.52** | 44 |

2. **Specialized Model** Specialized models fine-tuned on remote sensing data, including CCExpert (7B) , TEOChat (7B) , and Ours (7B) , exhibit mixed outcomes. Ours (7B) achieves 58.52% ST5-SCS through targeted training on RSCC, outperforming general models like Qwen2-VL (7B) . In contrast, CCExpert and TEOChat underperform in completeness and accuracy despite their domain focus, highlighting challenges in handling complex spatiotemporal reasoning. Proprietary models like Pixtral (12B) and InternVL 3 (8B) set performance benchmarks, while general models like BLIP-3 (3B) struggle with excessive output length (Avg_L=456 ) and low ROUGE scores (4.53% ).

3. **Repetition Issue** BLIP-3 and LLaVA-OneVision are prone to generative repetitive outputs. It is assumed that these models fail in dealing with remote sensing images or following complex instructions. This degeneration problem may be alleviated by switching decoding methods (e.g., Contrastive Decoding [67]) as well as adapting generation configurations [78].

### 4.3 Human Preference Study

While the language metrics can be biased, we ask experts to vote the best caption from two anonymous model output given the bi-temporal image pairs along with human labeled building damage masks from xBD dataset [21]. Results (Figure 4) reveal QvQ-Max (ground truth change captions) consistently outperformed all baselines, achieving win rates ranging from 80.7% (against InternVL3) to 99.0% (against CCExpert). While strong baselines like InternVL3 (**19.3%** wins) and mid-tier models (e.g., Pixtral [**14.6%**], Kimi-VL [**12.8%**]) showed moderate performance, our captions demonstrated superior accuracy in capturing fine-grained environmental changes critical for disaster response. Weak-performing multimodal baselines (LLaVA-Interleave [**5.2%**], Phi-4-MM [**4.9%**]) highlighted limitations in handling complex spatiotemporal reasoning, suggesting QvQ-Max's quantization-aware training and dynamic context adaptation mechanisms enhance generalization. These findings validate QvQ-Max as a state-of-the-art solution for vision-language tasks in remote sensing.

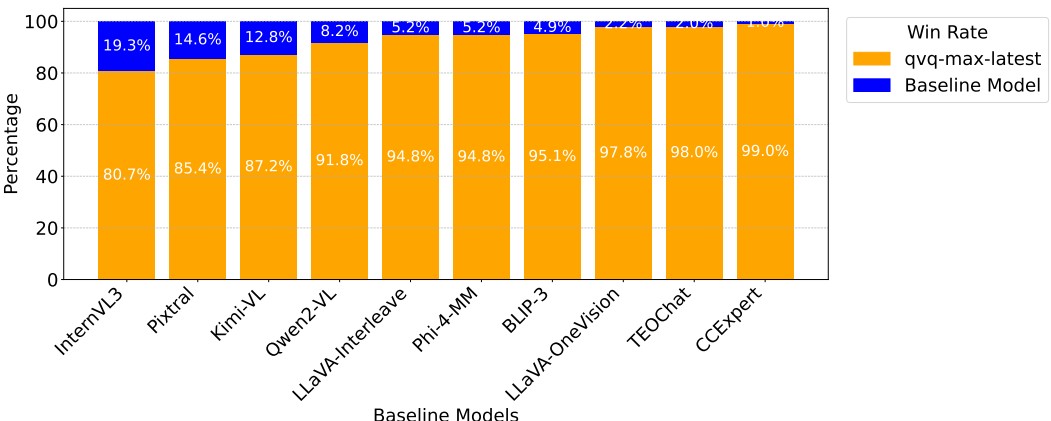

Figure 4: Win-rate from QvQ-Max (ground truth) to all baseline models on RSCC subset.

## 4.4    Inference-Time Augmentation

### 4.4.1    Employ Building Damage Info

Change caption result quality boost via augmentation with building damage info (Figure 5). It is witnessed that **auxiliary building damage info augmentation greatly improve the quality of change captions. We also find performance gets saturated equipped with auxiliary info regardless model size** (see quantitative results in Table 3 in Appendix A).

### 4.4.2    Scaling Correction Decoding

To investigate the effectiveness of scaling correction decoding strategies (e.g., VCD [31], DoLa [13] and DeCo [71]) in mitigating hallucinations during remote sensing change captioning, we evaluated their impact across varying model sizes for Qwen2.5-VL and InternVL3 (Figure 6). These strategies aim to align model outputs with input scale or context, reducing inconsistencies in multimodal reasoning.

For Qwen2.5-VL, Zero-Shot decoding achieves the highest similarity scores at smaller model sizes (3B–7B), while DeCo gradually closes the gap at larger scales (32B–72B). Notably, DoLa and VCD underperform across all sizes, suggesting limited utility for complex spatiotemporal reasoning tasks. In contrast, InternVL3 shows Zero-Shot decoding as the most consistent strategy, outperforming alternatives except at 14B, where DeCo marginally surpasses it. However, even at 38B parameters, scaling correction methods fail to achieve substantial gains over baseline performance. **We found no obvious boost using training-free correction decoding strategies on remote sensing change captioning task where we contribute to the abilities of complex visual reasoning instead of naive object level detection** (see Figure 7 for a case study).

## 5    Related Work

### 5.1    Remote Sensing Change Captioning Models

The remote sensing image change captioning (RSICC) task aims to generate detailed and accurate natural language to describe geospatial feature changes for remote sensing images captured at different times [9, 24]. Liu et al. [37] introduced RSICCformer, a Transformer-based approach that incorporates multiple cross-encoding modules to leverage differential features, focusing attention on the changing regions within each image. Similarly, Chg2Cap [10] presents a Transformer-based caption generation model that translates the relationship between image embeddings and word embeddings into descriptive text. Recent researches tend to use pre-trained LLM for language generation. GeoLLaVA [15] use fine-tune pretrained temporal MLLMs (i.e. Video-LLaVA [35] and LLaVA-NeXT-Video [33]) for detecting temporal changes in geographical landscapes. CCExpert [76], which is developed based on LLaVA-OneVision [32], introduces a difference-focused integration component. This module is designed to identify multi-scale variations between bi-temporal images

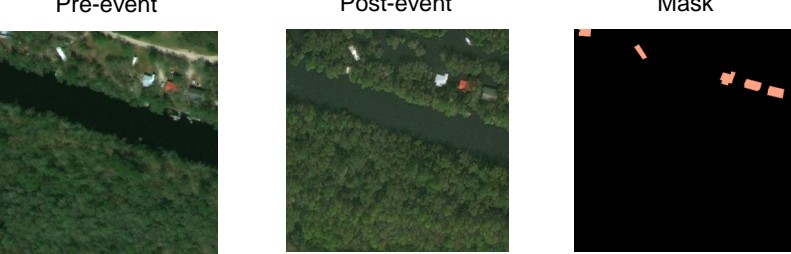

| Pre-event | Post-event | Mask |

**GT:** In a recent comparison of satellite imagery, significant changes have been observed, the water level has risen notably, submerging parts of the land previously visible, altering the landscape's appearance dramatically.

**InternVL3 8B (Zero-shot):** The satellite images reveal a slight change in the landscape near the river. A new structure with a red roof has appeared on the right side of the image, adjacent to the existing buildings. The road leading to the buildings appears to have been slightly widened. The surrounding vegetation remains consistent, with no significant changes in the forested area.

**InternVL3 8B (Textual Prompt):** A devastating storm has swept through the area, causing widespread destruction. The once undisturbed landscape now shows six buildings that have suffered major damage, with partial walls and roofs collapsed. The surrounding vegetation and structures are encroached upon by encroaching water, indicating a significant impact from flooding. The scene is one of desolation, with no signs of minor damage or intact structures, highlighting the severe nature of the storm's aftermath.

**InternVL3 8B (Visual Prompt):** A recent natural disaster has struck the area, causing significant destruction. Six buildings, previously standing undisturbed, now lie at varying levels of damage. The disaster, classified as Level 3, has left structures completely collapsed and surrounded by encroaching water, rendering them uninhabitable. The once serene landscape is now marked by the aftermath of the storm, with no signs of the former undamaged state. The extent of the devastation is evident, with no buildings remaining unaffected, highlighting the severe impact on the local area.

**InternVL 78B (Visual Prompt):** In a dramatic shift from the pre-event image, all six buildings now exhibit signs of major damage, indicated by orange coloring, revealing partial wall or roof collapses, significant water encroachment, and extensive structural compromise, as observed in the post-event satellite imagery. The once undisturbed area has been transformed into a scene of widespread devastation, with no structures escaping the impact of the storm, highlighting the severity of the natural disaster's reach.

Figure 5: Prompt augmentation results on RSCC (xBD: HURRICANE-FLORENCE). Critical descriptions are colored in green while incorrect and hallucinated sentences/words are red.

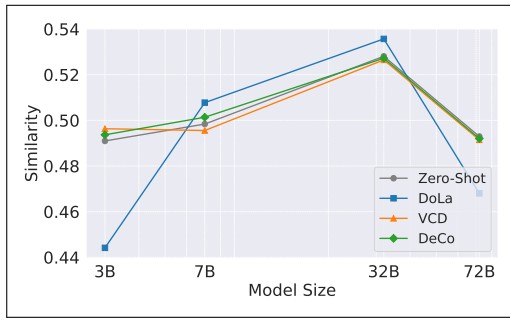

(a) Scaling correction decoding for Qwen2.5-VL

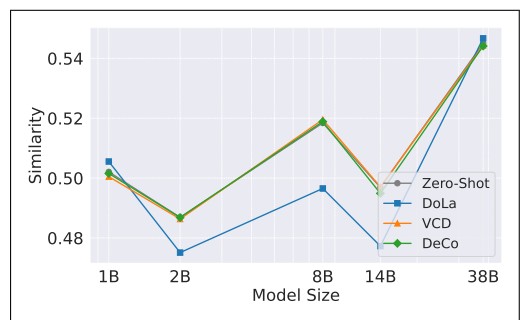

(b) Scaling correction decoding for InternVL3

Figure 6: Comparison of scaling correction decoding

and merge them into the initial image context. TEOChat [27] applies a shared vision-encoder to agument the temporal understanding capability of LLaVA-1.5 [40]. Despite the common architecture that intergrates a pre-trained LLM backbone and a vision encoder, Diffusion-RSCC [80] utilize a probabilistic diffusion model for RSICC that focus on pixel-level differences under long time span.

| Pre-event | Post-event | Mask |
|:---:|:---:|:---:|

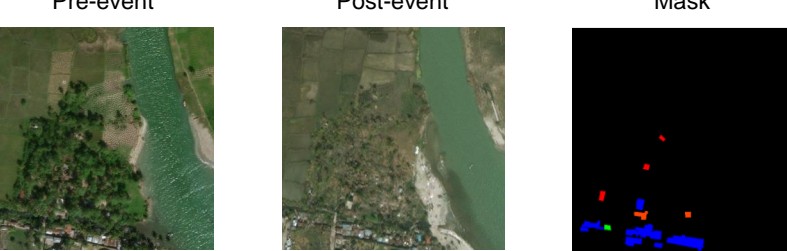

**GT**: A severe storm caused widespread destruction, transforming the landscape from pristine farmland into a scene of devastation. Of the 30 buildings assessed, 24 now exhibit minor damage—evident through partial roof losses, water encroachment, and scattered debris. Two structures suffered major damage, with walls collapsing and mud inundating interiors. Three buildings were completely destroyed, reduced to scorched remnants or submerged under floodwaters. Only one structure remained undamaged, standing resilient amid the chaos. The once-calm river overflowed, submerging fields and reshaping the shoreline, underscoring the storm's catastrophic force.

**Qwen2.5-VL 7B (Zero-shot)**: In the recent satellite imagery, a noticeable shift is observed in the vegetation cover along the riverbank, where the greenery appears more dense and vibrant compared to the previous image, suggesting potential reforestation or natural regrowth. Additionally, there seems to be an increase in the number of small structures near the water's edge, indicating possible human activity or settlement expansion. The overall landscape retains its agricultural patterns but exhibits subtle changes in land use and vegetation density.

**Qwen2.5-VL 7B (VCD)**: In the first satellite image, a dense cluster of trees is prominently visible near the riverbank, while in the second image, this area appears sparser, indicating a reduction in vegetation cover. Additionally, there seems to be a noticeable increase in the number of small structures or buildings along the river's edge in the second image, suggesting human activity or development.

**Qwen2.5-VL 7B (DoLa)**: A close look at the two satellite images reveals a significant shift in the greenery, with a dense, dark forest in the left image, while the right image shows a sparsely vegetated area, indicating a deforestation or land-use change, possibly for agricultural or urban development, with a visible increase in the open, arid landscape, and a new, small, man-made structure, hinting at human activity, in the bottom left corner, not present in the left image, further emphasizing the transformation.

**Qwen2.5-VL 7B (DeCo)**: In the recent satellite imagery, a noticeable shift is observed in the vegetation cover along the riverbank, where the greenery appears more dense and vibrant compared to the previous image. Additionally, there seems to be an increase in the number of small structures or buildings near the water's edge, indicating possible human habitation or development. The agricultural fields remain consistent, but their appearance suggests a different stage of crop growth or maintenance.

Figure 7: Correction decoding results on RSCC (xBD: HURRICANE-MATTHEW).

## 5.2 Remote Sensing Change Caption Dataset

Datasets that combine temporal and vision-language elements play a crucial role in training models to comprehend and merge temporal dynamics with linguistic information [39]. Unlike the VQA dataset [27, 15] that can be easily formatted through mask labels, the common practice of remote sensing change caption dataset is to further annotate existing change detection dataset with 5 sentences each image pairs, such as Dubai-CCD [25], LEVIR-CC [38] and WHU-CDC [65]. Given the generalization capability of commercial MLLMs, Wang *et al.* [76] leveraged GPT-4 [52], using the explicit information provided by the change masks to generate detailed change descriptions.

## 6 Limitations and Discussions

Due to the lack of proficient labels and complexity of image pairs themselves, the generated captions may contains vague descriptions which is even hard for experts to clarify. Besides, we only employ text similarity metrics because existing image-to-text captioning metrics (e.g., FLEUR [30], SPARC [28] and G-VEval [68]) only focus on single image which fail in multi-image scenario. We leave these parts for future work.

Also, our preliminary study have tested baseline VLMs on change detection and multi-label classification upon RSCC, where it show that naively employing VLMs such tasks yields much inferior results compared to specialized models [83, 65, 14]. Moreover, a recent paper [17] denotes that VLMs'

performance would degrade compared to their visual encoders only. Thus, we generally recommend using specialized models for such visual-centric tasks, and we hope the community will develop strong VLMs that are able to naively deal with these tasks.

# 7 Conclusions

In this work, we introduced RSCC, a large-scale event-driven remote sensing change caption dataset for disaster-awareness bi-temporal remote sensing image understanding. By leveraging visual reasoning model QvQ-Max, 62,351 pairs of pre-event and post-event images are annotated with detailed change caption. Furthermore, We established a comprehensive benchmark to facilitate the evaluation and development of large vision-language models in remote sensing change captioning. Our work focuses on promoting the training and evaluation of vision-language models for tasks related to understanding temporal remote sensing images.

# 8 Acknowledgements

We gratefully acknowledge the valuable suggestions and preliminary work contributed by student members Tesi Lin and Zeyu Zhang. We also thank Alibaba Cloud (Aliyun) for providing access to the Qwen API, which was instrumental in this project.

This work was supported by the National Natural Science Foundation of China under Grants 42394060 and 42394062 as well as the National Key Research and Development Program of China under Grant 2019YFE0127400.

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

# A Appendix

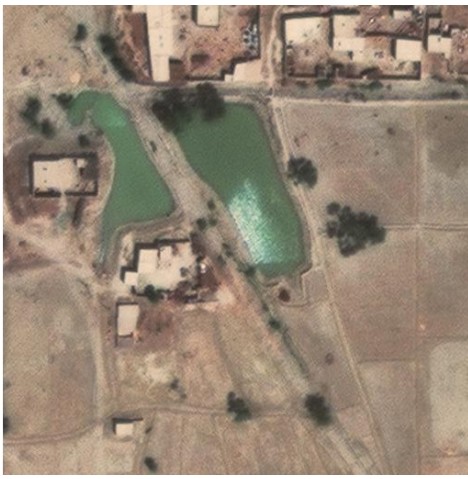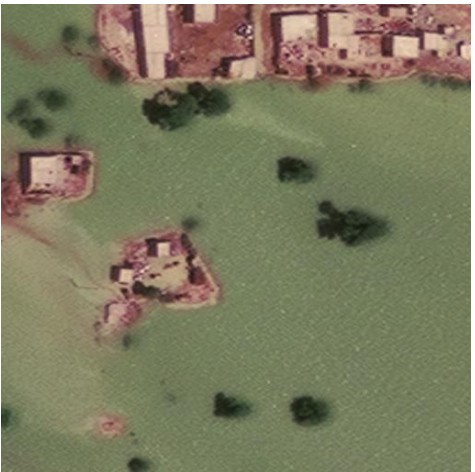

**GT (QvQ-Max):** A devastating flood has transformed the landscape, submerging vast agricultural fields and encroaching upon residential structures. The once-small water body has expanded dramatically, merging with adjacent farmlands and creating a sprawling inundated zone. Buildings near the water's edge now stand partially submerged, while the surrounding terrain shifts from parched earth tones to a dominant greenish-blue hue, signaling widespread flooding.

**Llama 4 Maverick:** Flooding has severely affected the area, with large portions of land now submerged underwater, surrounding buildings and altering the landscape.

**Pixtral-Large:** In a drastic transformation, what was once a landscape decorated with buildings and greenery is now largely submerged under floodwaters. The lower part of the area, previously home to structures and trees, is now overwhelmed, with only rooftops and tree crowns visible above the water. The extent of the submerged area emphasizes the severity of the flood, as water has inundated vast regions, leaving behind a stark contrast of water versus the remaining patches of land and standing structures.

**Gemini 2.5 Pro:** Satellite imagery starkly illustrates the impact of recent flooding on this community. Where dry land and distinct ponds once stood near buildings, widespread floodwaters now submerge the landscape. Structures appear isolated, surrounded by the extensive inundation that has overtaken the area.

**GPT 4.1:** Recent satellite images reveal the extensive impact of severe flooding in the region. What was once a landscape with distinct buildings, clear roads, and patches of vegetation is now submerged under floodwaters. Many structures are partially submerged, and open land has been transformed into a continuous waterbody, indicating significant disruption to homes and infrastructure.

**Qwen-VL-Max:** The satellite images reveal a stark transformation of the landscape following the flood. Previously dry fields are now submerged under water, with the waterline encroaching on nearby structures. The once visible roads and pathways are now obscured, indicating the severity of the flooding and its impact on the local infrastructure and daily life of the residents.

**Claude-3.7-Sonnet (thinking):** Devastating floods have transformed a rural community into a vast expanse of water, as seen in these striking satellite images. What was once dry land with scattered buildings and a small reservoir has become completely inundated, with structures now appearing as islands amid the swelling greenish floodwaters. Only rooftops and treetops remain visible above the waterline, highlighting the severe impact of this natural disaster on local infrastructure and livelihoods.

**o3 (thinking):** Satellite imagery captured before and after the disaster shows widespread inundation: small ponds have merged into a single expanse of water, submerging nearby fields and cutting off roads. Residential buildings that once stood on dry land now appear as isolated islands, with several rooftops partially underwater.

Figure 8: Comparisons of change captions of a pre-event image (Left) and a post-event image (Right) on RSCC (EBD: PAKISTAN-FLOODING) with Large Models . Words/sentences colored in green, red and purple denote to critical descriptions, incorrect descriptions and vague/undetermined descriptions respectively. Models with reasoning capabilities are underline.

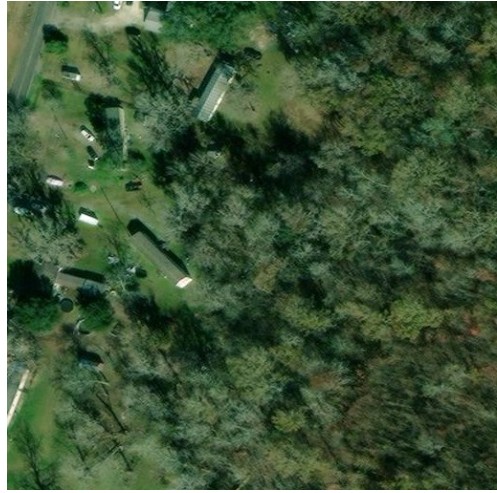 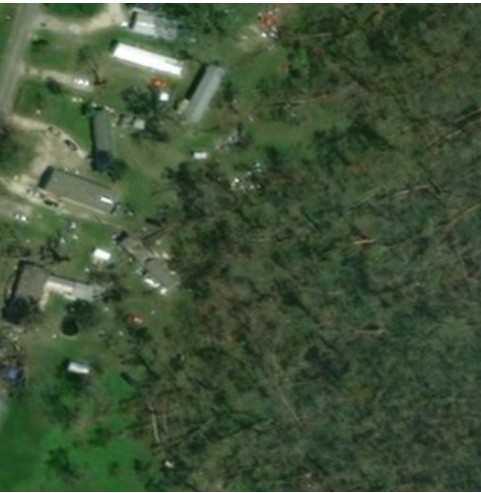

**GT (QvQ-Max):** The aftermath of the hurricane reveals significant devastation, with widespread deforestation evident as numerous trees lie uprooted or stripped bare, their branches scattered across the landscape. Debris litters the area, obscuring parts of the road and surrounding terrain, while several structures appear compromised, their roofs partially collapsed or missing entirely. The once lush greenery is now marred by patches of brown and gray, signaling the storm's brutal force.

**Llama 4 Maverick:** A satellite image shows extensive damage to buildings and trees in a rural area after a hurricane, with debris scattered across the landscape and significant disruption to the natural environment.

**Pixtral-Large:** Aerial views reveal a devastating transformation as the once tightly-knit and functional community has been scattered by the hurricane's might. Before the storm, the area was dotted with a dozen residential homes and commercial buildings, all surrounded by clearings and well-maintained roads. However, in the catastrophic aftermath, trees have been uprooted and scattered, infrastructures are significantly damaged, and lush green cover has receded, replaced by muddy trails and debris-strewn fields.

**Gemini 2.5 Pro:** Satellite imagery reveals significant damage following the hurricane. Numerous trees appear downed or damaged, particularly impacting the wooded area, and debris is scattered around structures, some of which show visible signs of damage.

**GPT 4.1:** The aftermath of the hurricane has brought significant changes to the area, as seen in the satellite images. Numerous trees have been stripped of their foliage or knocked down, leading to more visible ground patches and debris scattered across the landscape. Several structures and roads show signs of damage or disruption, and the transformation of the once lush area into a visibly affected zone highlights the severity and destructive impact of the hurricane.

**Qwen-VL-Max:** The area has suffered significant damage from the hurricane. Many structures appear to be destroyed or severely damaged, with debris scattered around. The forested area shows signs of defoliation and fallen trees, indicating the storm's intense wind force.

**Claude-3.7-Sonnet (thinking):** Devastating hurricane impact revealed in aerial imagery of rural community. The once-dense forest canopy appears significantly thinned, with numerous trees toppled throughout the woodland area. Several residential structures show visible damage, while others remain standing amid the destruction. The storm's powerful winds have transformed the landscape, leaving a patchwork of debris where continuous forest once stood. Emergency response teams are likely assessing damage to homes and infrastructure as recovery efforts begin in this hard-hit region.

**o3 (thinking):** Satellite imagery taken before and after hurricane shows extensive damage to the tree canopy east of the settlement, with large areas of forest flattened or uprooted. Several mobile homes and outbuildings remain standing, but roof damage and scattered debris are visible around the structures. Access roads stay passable, yet yards and clearings are now littered with fallen branches and storm-thrown materials, underscoring the storm's impact on this

Figure 9: Comparisons of change captions of a pre-event image (Left) and a post-event image (Right) on RSCC (EBD: HURRICANE-IDA) with Large Models. Words/sentences colored in green, red and purple denote to critical descriptions, incorrect descriptions and vague/undetermined descriptions respectively. Models with reasoning capabilities are underline.

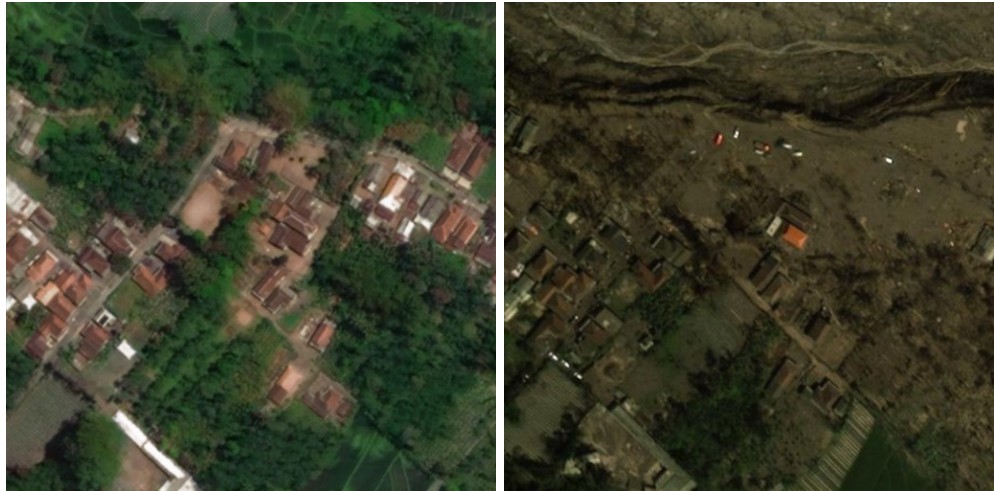

**GT (QvQ-Max):** A once-lush landscape transformed by volcanic fury: vibrant greenery and orderly settlements now lie buried under a thick blanket of ash and debris. The eruption has erased entire swaths of vegetation, leaving behind a desolate, gray terrain where homes and farmland once thrived. While some structures remain standing, their surroundings tell a stark tale of nature's destructive power.

**Llama 4 Maverick:** A satellite image reveals the devastating aftermath of a volcanic eruption, with widespread mud or ash covering the landscape and numerous buildings. The area has been significantly altered, with evidence of destruction and debris.

**Pixtral-Large:** In a dramatic turn of events, the satellite imagery reveals the devastating impact of a recent volcanic eruption on a once-thriving residential area. Previously, the region boasted numerous houses, lush greenery, and interconnected roads. However, the post-eruption image starkly contrasts with heavy ashfall covering the landscape, damaged or destroyed buildings, and a marked absence of vibrant foliage. The disaster has rendered much of the area unrecognizable, underscoring the volcano's ferocity and the urgent need for recovery efforts.

**Gemini 2.5 Pro:** Satellite imagery reveals the dramatic impact of the volcanic eruption, with previously lush areas and structures now buried under a thick layer of volcanic ash and debris flow. The landscape has been significantly altered, showcasing the destructive power of the event, particularly evident in the upper portion of the image where the flow path is clearly visible.

**GPT 4.1:** A devastating transformation is evident in the satellite images following the recent volcanic eruption. Previously lush and green landscapes, with clearly visible homes and farmlands, have been replaced by vast stretches of ash and volcanic debris. Many structures appear to be damaged or completely engulfed, with significant loss of vegetation and farmland now buried under volcanic material, underscoring the severe impact on the local community.

**Qwen-VL-Max:** The satellite images reveal the devastating impact of the recent volcanic eruption. Before the eruption, the area was characterized by a dense cluster of buildings and lush greenery. After the eruption, the landscape is now covered in a thick layer of ash and debris, with many structures destroyed or buried. The once vibrant community is now a desolate scene of destruction, highlighting the immense power and destructive force of the natural disaster.

**Claude-3.7-Sonnet (thinking):** Devastating scenes captured from satellite imagery reveal the catastrophic aftermath of a volcanic eruption that transformed a once-thriving community into a landscape of destruction. The before image shows a verdant settlement with buildings nestled among lush greenery, while the after image depicts a grim reality where volcanic flows have carved through the area, burying homes and infrastructure under thick layers of ash and mud. What was previously a vibrant green terrain has been dramatically altered to a barren gray-brown expanse, with only scattered structures remaining visible amidst the volcanic debris, highlighting the immense power and destructive capability of this natural disaster.

**o3 (thinking):** Satellite images reveal that the once verdant, village-lined landscape has been smothered by a sweeping blanket of volcanic ash and debris. Neighborhood blocks that previously showed orderly rows of red-tiled roofs and lush tree cover are now scarred: many buildings appear roofless or buried, vegetation is scorched brown, and roads are either obscured or severed by new ash channels. Farmland grids visible before the eruption have vanished beneath the dark flow, leaving only isolated structures and emergency vehicles dotting a barren expanse.

Figure 10: Comparisons of change captions of a pre-event image (Left) and a post-event image (Right) on RSCC (EBD: MOUNT-SEMERU-ERUPTION) with Large Models. Words/sentences colored in green, red and purple denote to critical descriptions, incorrect descriptions and vague/undetermined description respectively. Models with reasoning capabilities are underline.

(a)

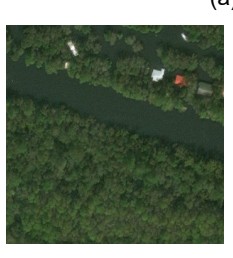

**Kimi-VL (3B):** The two satellite images depict a river flowing through a dense forest. In the first image, the river is bordered by a road on one side, with a few buildings and structures visible along the riverbank. The second image shows a similar scene, but the road and buildings are not visible, providing a clearer view of the forest and the river. The overall landscape in both images is characterized by lush greenery and a tranquil natural setting.

**GT:** In a recent comparison of satellite imagery, significant changes have been observed, the water level has risen notably, submerging parts of the land previously visible, altering the landscape's appearance dramatically.

------------------------------------------------------------------------------------------

(b)

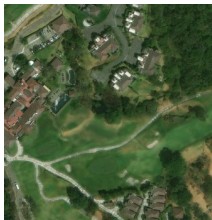 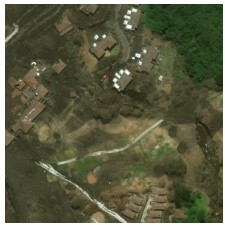

**BLIP-3 (4B):** The first satellite image shows a large body of water with a few small islands scattered around it. The second image shows the same area, but with a much larger and more detailed view of the islands and the surrounding water. The second image also shows more detail in the vegetation and terrain of the islands. (Repeat……)

**GT:** A dramatic transformation has occurred, the once lush green golf course now appears scorched and barren, the vibrant grass replaced by patches of dark earth, the clear pathways now marred by signs of destruction.

------------------------------------------------------------------------------------------

(c)

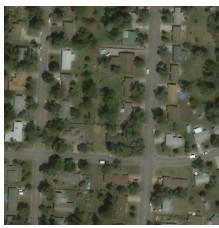 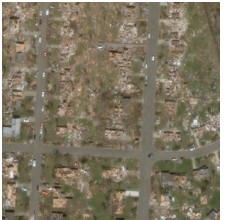

**Phi-4-Multimodal (4B):** The second image has more buildings.
**Qwen-2-VL (7B):** The image shows a significant change in the area, with many houses destroyed or damaged, and debris scattered across the streets.
**TEOChat (7B):** A hurricane has struck the area, resulting in significant damage.
**CCExpert (7B):** The area has been cleared of several buildings.

**GT:** A devastating transformation has occurred, as the once lush, green neighborhood is now replaced by a barren landscape of debris and destruction, highlighting the severe impact of a catastrophic event, with every structure reduced to rubble, and the streets eerily empty of life.

------------------------------------------------------------------------------------------

(d)

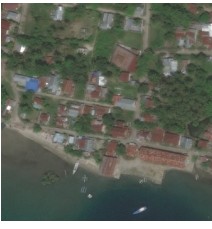 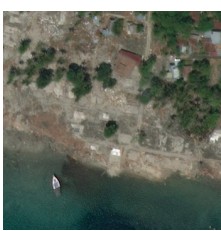

**LLaVA-Interleave (7B):** The first image shows a clear view of a coastal town with houses, roads, and green areas. The second image has been altered to show the town with a significant portion of the landscape covered in debris, suggesting a recent disaster or extensive destruction. The boat that was previously visible in the water is no longer seen in the second image.

**GT:** In a striking transformation, the once densely populated coastal area now appears significantly cleared, revealing patches of bare earth and fewer structures, while the waterway shows a reduction in visible boats, indicating a notable shift in human activity and environmental conditions.

Figure 11: Visualization of qualitative results. Critical descriptions are colored in green while incorrect and hallucinated sentences/words are red. More results are shown in Figure 12.

(e)

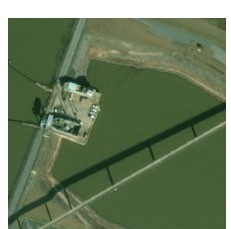 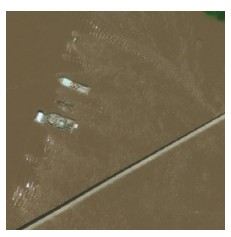

**LLaVA-OneVision (7B):** The satellite images show a significant change in the Earth's surface, with a large area affected by a natural disaster. The images depict a dramatic scene, with a wide area covered by a thick layer of debris. (Repeat…)

**GT:.** The area experienced a significant flood transformation, as evidenced by the pre-event image showing clear, undisturbed land with a single intact building, while the post-event image reveals the same location now submerged under murky floodwaters, with the building surrounded by water, indicating a shift from Disaster Level 0 to Disaster Level 2 conditions, highlighting the severe impact of the natural disaster on the infrastructure.

---------------------------------------------------------------------------------

(f)

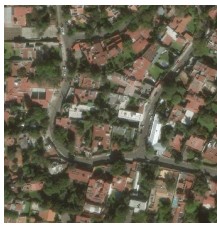 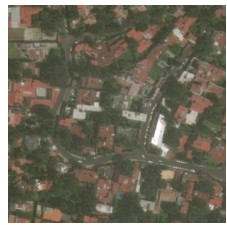

**InternVL 3 (8B):** The satellite images reveal a residential area with red-tiled roofs and greenery. In the second image, a white vehicle is visible on the road, which was not present in the first image. Additionally, there appears to be a small rectangular construction or alteration on the right side of the second image, not seen in the first image. The overall layout and density of buildings remain consistent between the two images.

**GT:** In an astonishing turn of events, the recent earthquake left the area virtually unscathed, as all 76 buildings remained intact, showing no signs of damage, cracks, or structural collapse, according to satellite imagery comparisons, highlighting the resilience of the structures and the community's preparedness. .

---------------------------------------------------------------------------------

(g)

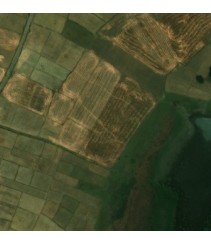 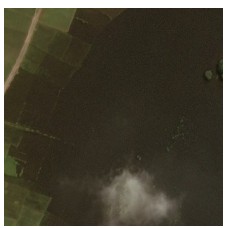

**Pixtral (12B):** In the provided satellite images, a significant change is observed in the agricultural fields. The first image shows fields with varying shades of green and brown, indicating different crop stages or types. A body of water is also visible near the center. In contrast, the second image reveals a more uniform color palette with predominantly brown and green fields, suggesting either a different season or crop rotation. The water body remains present but appears slightly altered in shape or size. These changes highlight the dynamic nature of agricultural landscapes over time.

**GT:** A severe flooding event has dramatically transformed the landscape, converting previously dry agricultural fields into vast expanses of waterlogged terrain, as evidenced by the stark contrast between the pre-event and post-event satellite imagery, highlighting the inundation of farmland, the disappearance of clear field boundaries, and the emergence of significant water bodies where none existed before.

Figure 12: More examples of RSCC. Critical descriptions are colored in green while incorrect and hallucinated sentences/words are red.

## A.1 RSCC Captioning Details

The experiments are implemented using the PyTorch framework and evaluated on an NVIDIA H800 GPUs (80GB). It takes about 1.1-8.3 seconds for captioning per image pair for all models with model size no more than 12B on a single H800 GPU.

We compare the performance of large-size MLLMs with zero-shot template (A.6) including open-source models such as Pixtral Large [48] and LLaMA-4 Maverick [45]. We also conduct case study on proprietary models including GPT-4.1 (2025-04-14) [50], Gemeni-2.5-Pro (2025-03-25) [18], and Qwen-VL Max (2025-01-25) [55], along with reasoning model such as Claude-3.7-Sonnet-Thinking (2025-02-25) [3] and o3 (2025-04-03) [53]. For results generation, We use default configurations of the above models. Figure 8, 9 and 10 show qualitative results of empirical study. We found

proprietary models outperform open-sourced models in completeness and accuracy. Visual reasoning notably improve the quality of caption in completeness but it also introduce vague information even hallucinations. As remote sensing change captioning requires world knowledge and complex reasoning, the latest state-of-the-art MLLMs seem to be insufficient.

## A.2 More Results

Figure 11 presents a qualitative comparison of vision-language models across diverse remote sensing scenarios, highlighting their ability to detect and describe change.

In Scenario (a), ground truth accurately identifies flooding as the disaster, highlights submergence of land, and links changes to water level rise, while Kimi-VL omits disaster causation and misrepresents structural disappearance as improved visibility.

In Scenario (b), ground truth accurately identifies the disaster type (fire/heat damage) and captures key changes: scorched vegetation, dark earth replacing greenery, and damaged pathways. Its description aligns with typical wildfire impacts (burnt surfaces, structural debris) while BLIP-3 incorrectly references a "body of water" and "islands," which are absent in the images, failing basic accuracy and relevance.

In Scenario (c), ground truth provides the most accurate, complete, and factually consistent description. It captures the catastrophic scale of destruction ("every structure reduced to rubble," "barren landscape"), explicitly mentions debris and empty streets, and aligns with typical patterns of severe wind-driven disasters (e.g., hurricanes or tornadoes). While it does not specify the disaster type, its focus on observable damage patterns (total structural collapse, vegetation loss) adheres strictly to visual evidence. Other captions either misinterpret the scene (Phi-4-MM, CCExpert), lack detail (TEOChat), or omit critical damage indicators (Qwen2-VL).

In Scenario (d), ground truth demonstrates superior completeness by explicitly mentioning "patches of bare earth," "fewer structures," and reduced boats, which align with visible changes in the images (e.g., exposed soil, collapsed buildings). While both captions lack explicit disaster type identification, ground truth's specificity on environmental and structural impacts ("significant clearing," "shift in human activity") enhances accuracy and clarity . LLaVA-Interleave's vague reference to "debris" and omission of key details (e.g., bare earth) makes it less precise. Both adhere to facts, but ground truth is richer detail elevates its overall quality.

Figure 12 shows more samples on RSCC subset along with baseline results. Table 3 shows overall quantitative results on RSCC subset. It is witnessed that auxiliary building damage info augmentation greatly improve the quality of change captions. We also find performance gets saturated equipped with auxiliary info regardless model size. We provide an additional metric BLEURT [6] [63], a learned evaluation metric to measure contextual similarity as well. However, the BLEURT is strongly biased on text length, which fails in valid evaluation. We are seeking for more reliable metrics in the future. Table 4 and  5 display RSCC data source details and baseline model configurations respectively.

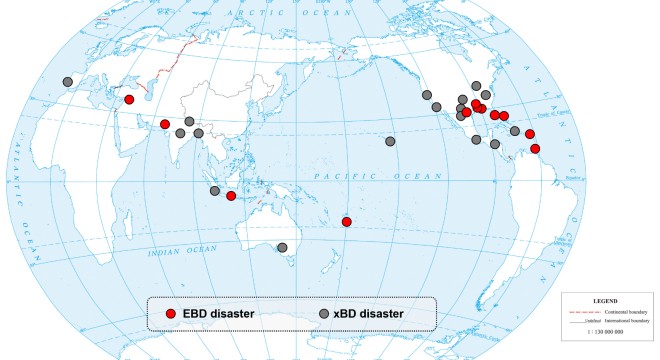

Figure 13: RSCC (EBD + xBD) distribution. Image Credit: Wang *et al.* [77].

---

[6] https://huggingface.co/lucadiliello/BLEURT-20-D12

Table 3: Detailed image caption performance on the subset of RSCC dataset. Avg_L denotes the average word number of generated captions. **Boldface** indicates the best performance while underline denotes the suboptimal performance.*We observe that BLIP-3 (XGen-MM) and LLaVA-OneVision tend to repeat their answer endlessly, which cause large caption lengths.

| Model (#activate params) | N-Gram | | Contextual Similarity | | Avg_L |
|---|---|---|---|---|---|
| | ROUGE(%)↑ | METEOR(%)↑ | BLEURT(%)↑ | ST5-SCS(%)↑ | |
| BLIP-3 (3B) [79] | 4.53 | 10.85 | 56.49 | 44.05 | *456 |
| + Textual Prompt | 10.07 (+5.54↑) | 20.69 (+9.84↑) | 56.79 (+0.30↑) | 63.67 (+19.62↑) | *302 |
| + Visual Prompt | 8.45 (-1.62↓) | 19.18 (-1.51↓) | 60.24 (+3.45↑) | 68.34 (+4.67↑) | *354 |
| Kimi-VL (3B) [29] | 12.47 | 16.95 | 45.11 | 51.35 | 87 |
| + Textual Prompt | 16.83 (+4.36↑) | 25.47 (+8.52↑) | 54.55 (+9.44↑) | 70.75 (+19.40↑) | 108 |
| + Visual Prompt | 16.83 (+0.00) | 25.39 (-0.08↓) | 54.24 (-0.31↓) | 69.97 (-0.78↓) | 109 |
| Phi-4-Multimodal (4B) [46] | 4.09 | 1.45 | 23.51 | 34.55 | 7 |
| + Textual Prompt | 17.08 (+13.00↑) | 19.70 (+18.25↑) | 52.00 (+28.49↑) | 67.62 (+33.07↑) | 75 |
| + Visual Prompt | 17.05 (-0.03↓) | 19.09 (-0.61↓) | 51.46 (-0.54↓) | 66.69 (-0.93↓) | 70 |
| Qwen2-VL (7B) [74] | 11.02 | 9.95 | 38.86 | 45.55 | 42 |
| + Textual Prompt | 19.04 (+8.02↑) | 25.20 (+15.25↑) | 52.64 (+13.78↑) | 72.65 (+27.10↑) | 84 |
| + Visual Prompt | 18.43 (-0.61↓) | 25.03 (-0.17↓) | 52.27 (-0.37↓) | 72.89 (+0.24↑) | 88 |
| LLaVA-NeXT-Interleave (8B) [33] | 12.51 | 13.29 | 42.80 | 46.99 | 57 |
| + Textual Prompt | 16.09 (+3.58↑) | 20.73 (+7.44↑) | 50.01 (+7.21↑) | 62.60 (+15.61↑) | 75 |
| + Visual Prompt | 15.76 (-0.33↓) | 21.17 (+0.44↑) | 50.08 (+0.07↑) | 65.75 (+3.15↑) | 88 |
| LLaVA-OneVision (8B) [32] | 8.40 | 10.97 | 46.27 | 46.15 | *221 |
| + Textual Prompt | 11.15 (+2.75↑) | 19.09 (+8.12↑) | **61.37** (+15.10↑) | 70.08 (+23.93↑) | *285 |
| + Visual Prompt | 10.68 (-0.47↓) | 18.27 (-0.82↓) | 60.59 (-0.78↓) | 69.34 (-0.74↓) | *290 |
| InternVL 3 (8B) [84] | 12.76 | 15.77 | 43.97 | 51.84 | 64 |
| + Textual Prompt | 19.81 (+7.05↑) | 28.51 (+12.74↑) | 56.51 (+12.54↑) | 78.57 (+26.73↑) | 81 |
| + Visual Prompt | 19.70 (-0.11↓) | 28.46 (-0.05↓) | 56.10 (-0.41↓) | **79.18** (+0.61↑) | 84 |
| Pixtral (12B) [47] | 12.34 | 15.94 | 43.74 | 49.36 | 70 |
| + Textual Prompt | **19.87** (+7.53↑) | **29.01** (+13.07↑) | 55.79 (+12.05↑) | 79.07 (+29.71↑) | 97 |
| + Visual Prompt | 19.03 (-0.84↓) | 28.44 (-0.57↓) | 54.99 (-0.80↓) | 78.71 (-0.36↓) | 102 |
| CCExpert (7B) [76] | 7.61 | 4.32 | 35.21 | 40.81 | 12 |
| + Textual Prompt | 8.71 (+1.10↑) | 5.35 (+1.03↑) | 39.01 (+3.80↑) | 47.13 (+6.32↑) | 14 |
| + Visual Prompt | 8.84 (+0.13↑) | 5.41 (+0.06↑) | 38.94 (-0.07↓) | 46.58 (-0.55↓) | 14 |
| TEOChat (7B) [27] | 7.86 | 5.77 | 39.47 | 52.64 | 15 |
| + Textual Prompt | 11.81 (+3.95↑) | 10.24 (+4.47↑) | 45.53 (+6.06↑) | 61.73 (+9.09↑) | 22 |
| + Visual Prompt | 11.55 (-0.26↓) | 10.04 (-0.20↓) | 45.31 (-0.22↓) | 62.53 (+0.80↑) | 22 |
| Ours (7B) | 14.99 | 16.05 | 45.50 | 58.52 | 44 |
| + Textual Prompt | 22.23 (+7.24↑) | 33.83 (+17.78↑) | 56.87 (+11.37↑) | 78.02 (+19.50↑) | 76 |
| + Visual Prompt | 22.37 (+0.14↑) | 33.81 (-0.02↓) | 57.02 (+0.15↑) | 78.87 (+0.85↑) | 79 |
| Qwen2.5-VL (72B) [58] | - | - | - | - | - |
| + Textual Prompt | - | - | - | 76.84 | 53 |
| + Visual Prompt | - | - | - | 76.85 | 57 |

Table 4: The 31 disaster events from RSCC dataset.

| Source | Disaster type | Disaster event | Event date |
|--------|---------------|----------------|------------|
| **xBD** | Earthquake | Mexico City earthquake | Sep 19, 2017 |
| | Wildfire | Portugal wildfires | Jun 17-24, 2017 |
| | Wildfire | Santa Rosa wildfires | Oct 8-31, 2017 |
| | Wildfire | Carr wildfire | Jul 23-Aug 30, 2018 |
| | Wildfire | Woolsey fire | Nov 9-28, 2018 |
| | Wildfire | Pinery fire | Nov 25-Dec 2, 2018 |
| | Volcano | Lower Puna volcanic eruption | May 23-Aug 14, 2018 |
| | Volcano | Guatemala Fuego volcanic eruption | Jun 3, 2018 |
| | Storm | Tuscaloosa, AL tornado | Apr 27, 2011 |
| | Storm | Joplin, MO tornado | May 22, 2011 |
| | Storm | Moore, OK tornado | May 20, 2013 |
| | Storm | Hurricane Matthew | Sep 28-Oct 10, 2016 |
| | Storm | Hurricane Florence | Sep 10-19, 2018 |
| | Flooding | Monsoon in Nepal, India, Bangladesh | Jul-Sep, 2017 |
| | Flooding | Hurricane Harvey | Aug 17-Sep 2, 2017 |
| | Flooding | Hurricane Michael | Oct 7-16, 2018 |
| | Flooding | Midwest US floods | Jan 3-May 31, 2019 |
| | Tsunami | Indonesia tsunami | Sep 18, 2018 |
| | Tsunami | Sunda Strait tsunami | Dec 22, 2018 |
| **EBD** | Hurricane | Hurricane Delta | Oct 8, 2020 |
| | Hurricane | Hurricane Dorian | Sep 1, 2019 |
| | Hurricane | Hurricane Ida | Oct 29, 2021 |
| | Hurricane | Hurricane Laura | Aug 26, 2020 |
| | Hurricane | Hurricane Irma | Sep 6, 2017 |
| | Hurricane | Hurricane Ian | Sep 26, 2022 |
| | Tornadoes | Texas Tornadoes | Mar 23, 2022 |
| | Volcanic Eruption | Mount Semeru Eruption | Dec 4, 2021 |
| | Volcanic Eruption | ST. Vincent Volcano | Apr 9, 2021 |
| | Volcanic Eruption | Tonga Volcano | Jan 15, 2022 |
| | Earthquake | Turkey Earthquake | Feb 6, 2023 |
| | Flooding | Pakistan Flooding | Jul 26, 2022 |

Table 5: Configuration of baseline models.

| Model Name | #Active Parameters | LLM | Image Encoder |
|------------|--------------------|-----|---------------|
| Kimi-VL | 3B | Moonlight-A3B-E18B | MoonViT |
| BLIP-3 | 4B | Phi-3-mini-4B | SigLIP |
| Phi-4-Multimodal | 4B | Phi-4-Mini 4B | SigLIP (LORA) |
| LLaVA-NeXT-Interleave | 7B | Qwen1.5 7B | SigLIP |
| Qwen2-VL | 7B | Qwen2-7B | DFN's ViT-H |
| LLaVa-OneVision | 7B | Qwen2 7B | SigLIP |
| InternVL 3 | 8B | Qwen2.5-7B | InternViT-300M |
| Pixtral | 12B | Mistral-Nemo-12B | PixtralViT |
| TEOChat | 7B | Vicuna-v1.5-7B | OpenCLIP-L/14 |
| CCExpert | 7B | Qwen2-7B | SigLIP |

### A.3 Access to Data

The RSCC dataset can be accessed and downloaded through our dedicated platform, which provides detailed views of the dataset components and their annotations. For practical examples and to download the dataset, visit our Huggingface repository (`https://huggingface.co/BiliSakura/RSCC`). Detailed metadata for the dataset is documented using the Croissant metadata framework, ensuring comprehensive coverage and compliance with the MLCommons Croissant standards, check [metadata](`https://huggingface.co/api/datasets/BiliSakura/RSCC`). Please check our Huggingface repo for metadata details. We also release our specialized model RSCCM (`https://huggingface.co/api/models/BiliSakura/RSCCM`).

### A.4 Author Statement and Data License

**Author Responsibility Statement:** The authors bear all responsibilities in case of any violations of rights or ethical concerns regarding the RSCC dataset.
**Data License Confirmation:** The dataset is released under the [CC-BY-4.0], which permits unrestricted use, distribution, and reproduction in any medium, provided the original work is properly cited.

### A.5 Broader Impacts

The dataset consists of non-sensitive, publicly available satellite images where no individual person or private property can be identified. Users are encouraged to use RSCC responsibly and ethically, particularly when developing applications that might impact environmental monitoring and urban planning.

### A.6 Prompt Template

---

**Prompt Template (Post Correction based on metadata)**

You will be provided with a change caption of a pair of remote sensing images, and metadata containing building damage statistics and disaster type. Perform the following analysis:

1. **Disaster Type Inference:**
   - Determine the disaster type (e.g., flood, wildfire) based on textual context.

2. **Keyword Evaluation:**
   - Extract disaster-relevant keywords from the caption.
   - Ensure these keywords are logically consistent with the inferred disaster type.

3. **Damage Statistics Validation:**
   - **Counts:** Compare the number of buildings per damage level in the caption (e.g., "24 minor-damaged") with the metadata values (e.g., {"minor-damage": 24}).
   - **Levels:** Verify that damage level terms (e.g., "destroyed" vs. "major-damage") match the metadata's labeling scheme.

4. **Flag Mismatches:**
   - **Keyword Mismatch:** Keywords incompatible with the disaster type (e.g., "volcanic ash" in a flood caption).
   - **Count Mismatch:** Discrepancies between caption and metadata (e.g., "24 minor-damaged" vs. metadata {"minor-damage": 20}).
   - **Level Mismatch:** Incorrect damage level terminology (e.g., "severe" instead of "major-damage").

5. **Return:**
   - "PASS" if all criteria are met.
   - "FAIL" with specific violations (e.g., "Count mismatch: Minor-damaged (caption:24 vs. metadata:20); Level mismatch: 'severe' instead of 'major-damage' ").

---

## Prompt Templates

### Prompt Template (Zero-Shot)

<image>\n<image>\n Give change description between two satellite images. Output answer in a news style with a few sentences using precise phrases separated by commas.

### Prompt Template (Textual Prompt)

<image>\n<image>\nThese two satellite images show a {disaster_type} natural disaster. Here is the disaster level descriptions:

- Disaster Level 0 (No Damage): Undisturbed. No sign of water, structural or shingle damage, or burn marks.
- Disaster Level 1 (Minor Damage): Building partially burnt, water surrounding structure, volcanic flow nearby, roof elements missing, or visible cracks.
- Disaster Level 2 (Major Damage): Partial wall or roof collapse, encroaching volcanic flow, or surrounded by water/mud.
- Disaster Level 3 (Destroyed): Scorched, completely collapsed, partially/completely covered with water/mud, or otherwise no longer present.

We already know that there are {number[all]} buildings. {number[no-damage]} buildings are no damaged. {number[minor-damage]} buildings are minor damaged, {number[major-damage]} building are major damaged, {number[destroyed]} buildings are destroyed. {number[unclassified]} buildings damage are unknown due to some reasons. Now, describe the changes that occurred between the pre-event and post-event images in a news style with the given disaster level descriptions.

### Prompt Template (Visual Prompt)

<image>\n<image>\nThese two satellite images show a {disaster_type} natural disaster. Here is the disaster level descriptions:

- Disaster Level 0 (No Damage): Undisturbed. No sign of water, structural or shingle damage, or burn marks.
- Disaster Level 1 (Minor Damage): Building partially burnt, water surrounding structure, volcanic flow nearby, roof elements missing, or visible cracks.
- Disaster Level 2 (Major Damage): Partial wall or roof collapse, encroaching volcanic flow, or surrounded by water/mud.
- Disaster Level 3 (Destroyed): Scorched, completely collapsed, partially/completely covered with water/mud, or otherwise no longer present.

We already know that there are {number[all]} buildings. {number[no-damage]} buildings are no damaged colored in green. {number[minor-damage]} buildings are minor damaged colored in blue, {number[major-damage]} building are major damaged colored in orange, {number[destroyed]} buildings are destroyed colored in red. {number[unclassified]} buildings damage are unknown due to some reasons colored in      . Now, describe the changes that occurred between the pre-event and post-event images in a news style with the given disaster level descriptions.

## A.7   Details of Human Preference Study

---

**Human Preference Guidelines**

You will be provided with 2 satellite images of the same area before and after a natural disaster event. Your task is to evaluate change captions generated by different vision language models and select the best one.
Evaluation Criteria:

1. Accuracy - Correct interpretation of damage patterns and disaster type
2. Completeness - Inclusion of relevant details (structures affected, disaster indicators)
3. Clarity - Clear, concise description without contradictions
4. Adherence to Facts - Consistency with typical disaster damage level

Follow the criteria and choose the best change caption by click the corresponding radio button.

---

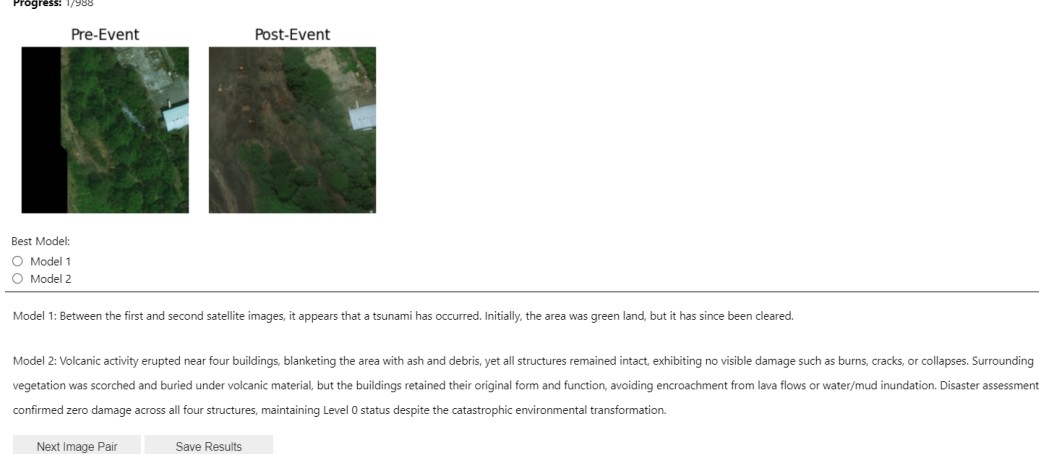

Figure 14: A screenshot of human preference study labeling interface.