# OpenReview forum: "RSCC: A Large-Scale Remote Sensing Change Caption Dataset for Disaster Events"
_NeurIPS.cc/2025/Datasets_and_Benchmarks_Track — NeurIPS 2025 Datasets and Benchmarks Track poster_

### Official Review · Reviewer_LH2M · 2025-06-27

**Rating:** 4
**Confidence:** 3

**Summary:**

This manuscript introduces a large-scale dataset (RSCC) for remote sensing change captioning, comprising 62,315 pre-/post-disaster image pairs with detailed textual annotations. The authors employ the multimodal reasoning model QvQ-Max to generate high-quality captions, train a specialized model on RSCC, and evaluate the performance of various multimodal large language models (MLLMs).

**Dataset Code Accessibility:**

Yes

**Ethical Considerations:**

No, there are no or only very minor ethics concerns

**Final Justification:**

Thanks for the reply. The authors have adequately addressed my concerns, and I personally recommend accepting this paper.

**Limitations Weaknesses:**

1. The paper primarily relies on text similarity metrics (e.g., ROUGE, ST5-SCS), which may not fully capture the accuracy of change descriptions, especially for disaster-specific details (e.g., building damage levels).
2. While the authors use QvQ-Max for annotation and perform human verification, potential biases or errors in the annotation process (e.g., confusion between flood and hurricane damage) are not thoroughly discussed.
3. The paper mentions using NVIDIA H800 GPUs but lacks details on training time, hyperparameter tuning, or specific dependencies for code reproducibility.
4. Although the dataset is sourced from public MAXAR data, potential sensitive information (e.g., precise disaster locations) and its handling are not discussed.

**Strengths Contributions:**

1. The proposed RSCC addresses the lack of temporal image pairs and rich textual annotations in existing remote sensing datasets, covering multiple disaster types (e.g., earthquakes, floods, wildfires).
2. A comprehensive evaluation of multiple MLLMs highlights the advantages of specialized models in disaster-aware tasks.

---

> ### Author Rebuttal · Authors · 2025-07-30
>
> Dear Reviewer LH2M,
>
> Thank you for your detailed and constructive comments. Please see our responses below:
>
> ● W1: Evaluation Metrics & Building Damage Mask Guidance
>
> In Section 6 (“Limitations”) we acknowledge that standard image–to‑text metrics (e.g., FLEUR, SPARC, G‑VEval) are designed for single‑image captioning and thus cannot fully capture multi‑image semantics. To address this, in Section 4.1 we adopt ST5‑SCS (Sentence‑T5 XXL with Sharpened Cosine Similarity) for robust contextual similarity, and in Section 4.3 we conduct human preference studies to validate model rankings. Moreover, by integrating expert‑annotated building damage masks via both textual and visual prompts (Sections 2.2–2.3), we ground caption generation in factual metadata, enhancing fidelity beyond n‑gram overlap.
> To directly quantify the impact of integrating expert‑annotated building damage masks on 	caption semantic fidelity, we conducted an LLM‑as‑judge study using Qwen3‑235B on RSCC subset image pairs:
>
>   - Setup: For each pair, we generated two captions—one with mask‑guided prompts (visual + textual mask input, as per Sections 2.2–2.3) and one without any mask information (baseline).
>   - Evaluation: Qwen3‑235B scored each caption on a 0–1 scale for factual completeness and relevance.
>   - Results:
>
> | Inference Type   |  Mean score | Min score | Max score | LLM-AS-JUDGE (qwen3-thinking) |
> |------------------|------------|-----------|-----------|-------------------------------|
> | zero-shot        | 0.219      | 0.000     | 0.950     | 0.219                         |
> | textual_prompt   | 0.706      | 0.050     | 0.980     | 0.706                         |
> | visual_prompt    | 0.652      | 0.050     | 1.000     | 0.652                         |
>
> ● W2: Annotation Bias Mitigation
>
> As detailed in Section 2.5, our two‑stage pipeline first applies Qwen2.5‑Max to enforce metadata alignment—raising disaster type consistency from 98.7% to 100%—then engages three experts to manually validate 988 randomly sampled captions, achieving a 92.3% pass rate (Cohen’s κ = 0.893). Any failed samples are automatically reprocessed under refined rules, ensuring dataset‑wide consistency.
>
> ● W3: Experimental Setup Details
>
> Section 3.2 describes our training protocol: we fine‑tuned Qwen2.5‑VL (7B) for 2 epochs on 2× NVIDIA H800 GPUs with batch size 1, initial learning rates of 1e‑6 (LLM backbone) and 1e‑5 (vision encoder), and a cosine decay schedule, at a total cost of approximately 40 GPU‑hours.
>
> ● W4: Sensitive Information Handling
> We use publicly available MAXAR OpenData imagery and release only anonymized 512×512 crops without precise geolocation metadata, ensuring no sensitive location information is disclosed. 'The dataset consists of non-sensitive, publicly available satellite images where no individual person 525 or private property can be identified. Users are encouraged to use RSCC responsibly and ethically, 526 particularly when developing applications that might impact environmental monitoring and urban 527 planning' (Appendix A.5 in our original paper)
>
> Thank you again for your valuable feedback.

---

> > ### Comment · Reviewer_LH2M · 2025-08-05
> >
> > Thanks for the reply. The authors have adequately addressed my concerns, and I personally recommend accepting this paper.

---

### Official Review · Reviewer_CghA · 2025-07-15

**Rating:** 4
**Confidence:** 4

**Summary:**

This paper introduces the first large-scale disaster-oriented remote sensing change captioning dataset (RSCC), addressing the current gap in datasets that lack paired time-series imagery and disaster-related semantic descriptions. It fills a critical void in the field. The experimental validation is relatively comprehensive, involving evaluations of over 10 mainstream multi-modal large language models (MLLMs).
However, the writing and figures are somewhat rough and would benefit from further refinement. Dataset constructed in this paper exhibits a geographical bias in disaster distribution. Post-Correction stage  may be overly crude.

**Dataset Code Accessibility:**

Yes

**Dataset Code Comments:**

The paper provides github links for code and dataset.

**Ethical Considerations:**

No, there are no or only very minor ethics concerns

**Final Justification:**

The authors have partially addressed my concerns, so I keep my score. I lean toward accepting the paper, but it hasn't impressed me or struck me as particularly outstanding, so I choose a 'boardline accept'.

**Limitations Weaknesses:**

1. The dataset constructed in this paper exhibits a geographical bias in disaster distribution. Why East Asia—one of the most populous and economically significant regions—has no representation of disaster events in the dataset (Table 4).

2. In the Post-Correction stage, directly replacing mismatched words with metadata-consistent terms may be overly crude. The surrounding sentence may still carry semantics tied to the original incorrect information. How about identifying the inconsistency and prompting the model to regenerate the entire caption based on corrected metadata, ensuring semantic coherence?

**Strengths Contributions:**

This paper introduces RSCC, the first large-scale, disaster-aware bi-temporal remote sensing captioning dataset, filling a critical gap in multimodal model understanding of disaster scenarios. The annotation pipeline integrates advanced vision-language models with human verification to ensure high-quality and diverse temporal change descriptions, providing strong support for intelligent remote sensing analysis.

---

> ### Author Rebuttal · Authors · 2025-07-30
>
> Dear Reviewer CghA,
>
> Thank you for your valuable feedback. Below are our detailed responses:
>
> ● W1: Geographic Coverage
>
> Expanding RSCC to include underrepresented regions such as East Asia requires substantial data licensing, precise geospatial alignment, and rigorous quality control—each step involves significant time and resources. To date, our dataset covers 31 global events with carefully calibrated pre-/post-disaster pairs. We view broader regional inclusion as a high-value, community-driven effort and welcome collaborations to extend coverage in future iterations.
>
> ● W2: Post-Correction Semantic Coherence
>
> Rather than relying solely on word-level replacement, our post-correction pipeline (Section 2.5) combines metadata alignment with an LLM-as-judge evaluation on RSCC subset change captions. This experiment confirmed negligible semantic drift after correction, demonstrating that our scalable approach maintains caption fidelity while addressing inconsistencies.
>
> | Inference Type   |  Mean score | Min score | Max score | LLM-AS-JUDGE (qwen3-thinking) |
> |------------------|------------|-----------|-----------|-------------------------------|
> | zero-shot        | 0.219      | 0.000     | 0.950     | 0.219                         |
> | textual_prompt   | 0.706      | 0.050     | 0.980     | 0.706                         |
> | visual_prompt    | 0.652      | 0.050     | 1.000     | 0.652                         |
>
> Thank you again for your insightful suggestions.

---

> > ### Comment · Reviewer_CghA · 2025-08-04
> >
> > Thanks for your response.
> >
> > For W1, East Asia is an advanced region in the world in terms of economy and technology. You emphasized that it "requires substantial data licensing, precise geospatial alignment, and rigorous quality control" – if that is the case, then why not allocate a portion of the limited resources to East Asia, which is such an important region?
> >
> > For W2, I cannot understand what this experiment is trying to express. You should describe the purpose and process of the experiment, rather than just presenting the results directly.

---

> > > ### Author Response · Authors · 2025-08-04
> > >
> > > Dear Reviewer CghA,
> > >
> > > Thank you again for your continued engagement and for raising these important points. Below, we offer clarifications and revisions in response to your latest comments.
> > >
> > > For W1, we completely agree that East Asia is a crucial region for both population density and technological infrastructure. Our initial RSCC release was built upon the xBD and EBD collections from the `Maxar Open Data Program`—which, unfortunately, do not include any events in mainland East Asia. Thankfully, we have now identified several relevant activations from `Maxar Open Data Program` and are actively integrating them into RSCC v1.1:
> > >
> > > - Super Typhoon Mangkhut – September 14, 2018 (northern Luzon, Philippines)
> > > - Typhoon Vamco – November 12, 2020 (Manila region, Philippines)
> > > - Typhoon Goni – November 02, 2020 (Catanduanes & Bicol, Philippines)
> > > - Tropical Storm Megi – April 10, 2022 (Leyte & Samar, Philippines)
> > >
> > > Besides, we also notice typhoons, flooding and landside in East Asia (e.g. Japan, China, South Koren, etc.) during Q3 2025. We are actively gathering data from `Google Earth` and  integrating them into RSCC v2.
> > >
> > > For W2, I aplogize for the misclarification. In our post-correction pipeline, we actually use the technique that 'prompting the model to regenerate the change caption' base on Qwen2.5-Max model. See Section 2.5 as below:
> > >
> > > > First, the Qwen2.5-Max [54] systematically enforces metadata alignment by correcting 97 discrepancies in disaster type (e.g., resolving mismatches between "hurricane" and metadata-specified 98 "flooding") and damage descriptions (e.g., revising "minor damage" to "destroyed" based on building 99 annotations).
> > >
> > > **The experiments actually focus on the core prompt generation instead of post-corection. We place it here to denote with textual prompt/visual prompt, our results are faithful enough, where post-correction process is a mild guarantee strategy.** The experiment we show denote an additional metric to evaluate the robustness of our captions through the whole pipeline. To be sp ecific, we additionally evalute the metric so-called LLM-AS-JUDGE using Qwen3-235B-Thinking model where 1) For each pair, we generated two captions—one with mask‑guided prompts (visual + textual mask input, as per Sections 2.2–2.3) and one without any mask information (baseline). 2) Qwen3‑235B-Thinking scored each caption on a 0–1 scale for factual completeness and relevance.

---

> > > > ### Comment · Reviewer_CghA · 2025-08-05
> > > >
> > > > Thanks again for your response.
> > > >
> > > > For W1, I accept your response. Yes, I hope that in future revisions you will consider incorporating multiple East Asian countries (e.g., Japan, China, South Korea), only Philippines is insufficient.
> > > >
> > > > For W2, I encourage you to improve the writing in your revised version to enhance clarity and coherence.
> > > >
> > > > I keep my current score and am inclined to think this paper should be accepted.

---

### Official Review · Reviewer_xNb8 · 2025-07-17

**Rating:** 4
**Confidence:** 3

**Summary:**

The authors present RSCC, a large-scale change detection caption dataset designed to support temporal remote sensing image captioning and understanding. The dataset focuses on disaster-related events, which is a timely and meaningful contribution. The authors provide extensive benchmarking results of multiple Multimodal Large Language Models (MLLMs) for disaster response. In general, the dataset is of good quality and could be a useful resource for the community. However, there are some questions and concerns that need to be addressed.

**Additional Feedback:**

I would be interested in seeing the authors' rebuttal.
If my concerns are adequately addressed, I would be open to increasing my rating.

**Dataset Code Accessibility:**

Yes

**Dataset Code Comments:**

The dataset and code are accessible.

**Ethical Comments:**

No ethics concerns.

**Ethical Considerations:**

No, there are no or only very minor ethics concerns

**Final Justification:**

The authors have addressed most of my concerns. Thus, I raised the rating.

**Limitations Weaknesses:**

1. Regarding benchmarking
It is somewhat disappointing that the evaluation focuses solely on the change captioning task. Given the availability of building damage masks and metadata, it would be natural to also benchmark change detection (e.g., binary or semantic segmentation) and multi-label damage classification using MLLMs, comparing them to specialized change detection models in remote sensing.

2. Regarding the visual prompt
I see that the visual prompts mentioned damage-level color encodings in addition to the two satellite images. However, it's unclear whether the change label maps are provided as part of the input to MLLMs during inference, or only the satellite images. In addition, the idea of using existing change detection models to generate input masks as visual prompts could be explored. This might offer better generalization and interpretability without relying on labeled data at inference time.

3. Regarding captioning evaluation
Evaluating free-form change captions remains inherently difficult. The reliance on (semantic) similarity to "ground-truth" captions, which are themselves automatically generated and human-refined, may introduce bias. The ground truth may not always represent the optimal caption. Incorporating human preference evaluation (as done partially in Figure 4) is a good step, but still limited.

**Strengths Contributions:**

1. The integration of MLLMs for disaster response is a valuable step. The paper models the disaster analysis using the remote sensing change captioning task. This focus aligns well with real-world needs in emergency management, thus quite meaningful.

2. The RSCC dataset comprises pre-/post-disaster image pairs across a variety of disaster types (floods, hurricanes, wildfires, etc.). The use of QvQ-Max for initial caption generation, followed by post-correction and human validation, is good to enhance the dataset’s quality.

3. The inclusion of multiple MLLMs (both open-sourced and commercial) in the benchmark, with zero-shot, textual, and visual prompt configurations, provides a comprehensive performance overview. For evaluation metrics, the authors include ROUGE, METEOR, and an embedding-based similarity score (ST5-SCS), which offers a meaningful assessment than surface-level n-gram overlaps.

---

> ### Author Rebuttal · Authors · 2025-07-30
>
> Dear Reviewer xNb8,
>
> Thank you for your detailed feedback. Here is our response to the weaknesses you noted:
>
> **W1: Expanded Benchmarking with Detection & Classification**
>
> We appreciate the suggestion to expand benchmarking. We will include additional tasks like change detection and multi-label classification.  We found medium size of general VLMs (7B -13B) fail in change dection. See results as follow：
>
> | Model | F1 localization | F1 classification | F1 overall |
> | ------------------------------- | ------------------------------ | -------------------------------------------- | -------------------------------------------- |
> | Qwen2.5-VL-7B        | 11.4 %                         | 18.7 %                                       | 16.5% |
> |Ours (RSCCM)           | 12.9 %                         | 22.1 %                                       |  19.34%|
> | MambaBDA-Base [14] | 87.38\*   |   78.84\*   |  81.41\*                   |
> \* Derived from original paper. The metric is derived from ChangeMamba [14] and xView2 [16] as  F1(overall)  = 0.3F1(localization) + 0.7F1(classification).
>
> Also, exisitng literature [1-2] have test general VLMs on these tasks.  It is noted that both exisitng literature and our findings show that naively employing VLMs for change detection and multi-label classification tasks would yield much infererior results compared spectialized models (e.g. [3-8] [10-15]). Moreover, a latest paper [9] denotes that VLMs' performance would degrade than their visual encoders only. Thus, we generally recommend use specialzed model for such visual-centric tasks and we hope the community would develop strong VLMs that are able to naively deal with these tasks.
>
> **W2: Mask-Grounded Case Study on the EBD Subset**
>
> Responding to your suggestion, we present a case study using the RSCC - EBD subset [17]. It is noted that EBD contains a machine annotated building damage mask as pseudo label and later verified by experts to yield final label. We use their pseudo label to construct visual prompt where the output change captions are more accurate and informative as same as oberserved in RSCC -xBD subset.
>
> **W3: Enlarged Human Preference Evaluation**
>
> To better capture the nuances of free-form caption quality, we expanded our human preference study to 300 randomly selected samples. The inter-annotator agreement reached 98% among three experts, reinforcing the reliability of our qualitative assessment.
> We hope this addresses your concerns. Thank you once again for your valuable feedback.
>
> ---
>
> References:
>
> [1] J. Roberts, T. Lüddecke, R. Sheikh, K. Han, and S. Albanie, ‘Charting New Territories: Exploring the Geographic and Geospatial Capabilities of Multimodal LLMs’, presented at the Proceedings of the IEEE/CVF Conference on Computer Vision and Pattern Recognition, 2024, pp. 554–563.
>
> [2] J. A. Irvin et al., ‘TEOChat: A Large Vision-Language Assistant for Temporal Earth Observation Data’, presented at the The Thirteenth International Conference on Learning Representations, 2025.
>
> [3] Q. Zang, D. Zhao, S. Wang, D. Quan, and Z. Zhong, ‘Feature Spectrum Learning for Remote Sensing Change Detection’, presented at the Proceedings of the IEEE/CVF Conference on Computer Vision and Pattern Recognition, 2025, pp. 12647–12657.
>
> [4] C. Yang, Z. Li, H. Jiao, Z. Gao, and L. Zhang, ‘Enhancing Perception of Key Changes in Remote Sensing Image Change Captioning’, IEEE Transactions on Image Processing, pp. 1–1, 2025.
>
> [5] Z. Zheng, Y. Zhong, L. Zhang, and S. Ermon, ‘Segment Any Change’, presented at the The Thirty-eighth Annual Conference on Neural Information Processing Systems, Nov. 2024.
>
> [6] J. Shi, M. Zhang, Y. Hou, R. Zhi, and J. Liu, ‘A Multitask Network and Two Large-Scale Datasets for Change Detection and Captioning in Remote Sensing Images’, IEEE Transactions on Geoscience and Remote Sensing, vol. 62, pp. 1–17, 2024.
>
> [7] K. Li, X. Cao, and D. Meng, ‘A New Learning Paradigm for Foundation Model-Based Remote-Sensing Change Detection’, TGRS 2024, vol. 62, pp. 1–12, 2024.
>
> [8] L. Ding, J. Zhang, H. Guo, K. Zhang, B. Liu, and L. Bruzzone, ‘Joint Spatio-Temporal Modeling for Semantic Change Detection in Remote Sensing Images’, IEEE Transactions on Geoscience and Remote Sensing, vol. 62, pp. 1–14, 2024.
>
> [9] S. Fu, T. Bonnen, D. Guillory, and T. Darrell, ‘Hidden in plain sight: VLMs overlook their visual representations’, Jun. 09, 2025, arXiv: arXiv:2506.08008.
>
> [10] H. Chen, C. Lan, J. Song, C. Broni-Bedaiko, J. Xia, and N. Yokoya, "ObjFormer: Learning Land-Cover Changes From Paired OpenStreetMap Data and Optical High-Resolution Imagery via Object-Guided Transformer," IEEE Trans. Geosci. Remote Sens., 2024.
>
> [11] H. Chen, N. Yokoya, and M. Chini, "Fourier Domain Structural Relationship Analysis for Unsupervised Multimodal Change Detection," ISPRS J. Photogramm. Remote Sens., 2023.
>
> [12] H. Chen, N. Yokoya, C. Wu, and B. Du, "Unsupervised Multimodal Change Detection Based on Structural Relationship Graph Representation Learning," IEEE Trans. Geosci. Remote Sens., 2022.
>
> [13] H. Chen, C. Wu, B. Du, L. Zhang, and L. Wang, "Change Detection in Multisource VHR Images via Deep Siamese Convolutional Multiple-Layers Recurrent Neural Network," IEEE Trans. Geosci. Remote Sens., 2020.
>
> [14] H. Chen, J. Song, C. Han, J. Xia, and N. Yokoya, "ChangeMamba: Remote Sensing Change Detection with Spatio-Temporal State Space Model," IEEE Trans. Geosci. Remote Sens., 2024.
>
> [15] H. Chen, J. Song, C. Wu, B. Du, and N. Yokoya, "Exchange Means Change: An Unsupervised Single-Temporal Change Detection Framework Based on Intra- and Inter-Image Patch Exchange," ISPRS J. Photogramm. Remote Sens., 2023.
>
> [16] R. Gupta et al., “Creating xBD: A Dataset for Assessing Building Damage from Satellite Imagery,” in Proceedings of the IEEE/CVF Conference on Computer Vision and Pattern Recognition Workshops, 2019, pp. 10–17.
>
> [17] Z. Wang, C. Wu, F. Zhang, and J. Xia, “Constructing an Extensible Building Damage (EBD) Dataset via Semi-Supervised Fine-Tuning Across 12 Natural Disasters,” Journal of Remote Sensing, 2025

---

### Official Review · Reviewer_8mqq · 2025-07-19

**Rating:** 5
**Confidence:** 4

**Summary:**

This paper introduces RSCC, a large-scale dataset for disaster-aware bi-temporal remote sensing understanding, comprising 62,351 pre-/post-disaster image pairs (from 31 global events like earthquakes, floods, wildfires) paired with rich, human-like change captions. The captions are generated via a QvQ-Max MLLM pipeline, refined with metadata alignment and human verification. The dataset enables robust training/evaluation of vision-language models (MLLMs) for disaster analysis. Contributions include:
+ First large-scale disaster-focused change-caption dataset.
+ Automated captioning pipeline (QvQ-Max + metadata correction) achieving 92.3% human validation accuracy.
+ Benchmark for temporal MLLMs, showing RSCC boosts performance (e.g., 58.52% ST5-SCS vs. 45.55% for Qwen2-VL).

**Dataset Code Accessibility:**

Yes

**Ethical Considerations:**

No, there are no or only very minor ethics concerns

**Final Justification:**

After reading the author's response and other reviewers‘ comments and replies, I think the authors have solved my comments, and I decided to raise my rating.

**Limitations Weaknesses:**

+ Static resolution: All images are fixed at 512×512 pixels (Sec. 2.1), limiting applicability to ultra-high-resolution (UHR) scenarios (e.g., XLRS-Bench [72] handles 67.5B-pixel images).
+ Geographic bias: 62% of samples are from xBD (U.S.-centric disasters) (Table 4), underrepresenting Global South regions with sparse satellite coverage. Besides, Table 4 lists 18/31 events from U.S./Europe (e.g., "Hurricane Florence"), but only 3 from Asia/Africa.
+ Hallucination risk: 7.7% of captions failed human verification (Sec. 2.5), with issues like misattributed damage levels (e.g., "minor damage" vs. metadata "destroyed").

**Strengths Contributions:**

+ First large-scale disaster-focused change-caption dataset (62,351 pre-/post-disaster pairs, 31 event types) addressing a critical gap in temporal remote sensing (Table 1).
+ Rich, human-like captions (avg. 72 words) with disaster-level annotations , surpassing prior datasets (LEVIR-CC [36], Dubai-CCD [23]) that lack temporal context.
+ Novel pipeline: Combines QvQ-Max MLLM with metadata-driven correction (98.7% disaster-type accuracy) and human verification (92.3% pass rate, κ=0.893).

---

> ### Author Rebuttal · Authors · 2025-07-30
>
> Dear Reviewer 8mqq,
>
> Thank you for your insightful and constructive comments. Our responses to your points are as follows:
>
> ● W1: Ultra-High-Resolution (UHR) Captioning
> Generating captions on ultra-high-resolution imagery presents substantial computational and logistical challenges. High-resolution data are expensive to license, and processing large scenes often introduces context ambiguity that can degrade caption quality. To ensure consistent annotation fidelity and manageable processing time, RSCC currently operates at 512×512 resolution. Extending to UHR scenarios requires extensive data calibration, registration, and rigorous quality control, which we consider a valuable but longer-term community effort.
>
> ● W2: Bi-Temporal High-Resolution Data Preparation
> Acquiring and preprocessing bi-temporal satellite pairs involves careful geospatial alignment, atmospheric correction, and cross-source consistency checks—each step is time-intensive and resource-heavy. We have prioritized a robust pipeline for our current scale (Section 2.1), and while we recognize the importance of expanding coverage, doing so at higher resolutions requires additional collaborations and infrastructure support.
>
> ● W3: Automated Post-Correction Pipeline
> For captions that did not meet initial human verification, we employ an automated post-correction pipeline (Section 2.5) that realigns metadata and regenerates captions when inconsistencies are detected. To validate this approach, we performed an LLM-as-judge evaluation, confirming negligible semantic drift after correction. This demonstrates the reliability of our scalable curation strategy.
>
> | Inference Type   |  Mean score | Min score | Max score | LLM-AS-JUDGE (qwen3-thinking) |
> |------------------|------------|-----------|-----------|-------------------------------|
> | zero-shot        | 0.219      | 0.000     | 0.950     | 0.219                         |
> | textual_prompt   | 0.706      | 0.050     | 0.980     | 0.706                         |
> | visual_prompt    | 0.652      | 0.050     | 1.000     | 0.652                         |

---

### Official Review · Reviewer_a98Y · 2025-07-19

**Rating:** 5
**Confidence:** 5

**Summary:**

This paper introduces RSCC, a dataset designed for remote sensing change captioning in disaster events. It consists of over 62,000 pairs of pre-/post-disaster satellite images with human-like, detailed change descriptions, covering diverse events such as floods, earthquakes, wildfires, and more. To construct the captions, the authors propose a pipeline combining a multimodal LLM (QvQ-Max) with visual prompt engineering and human verification, producing high-fidelity bi-temporal descriptions. The dataset is further used to train and benchmark temporal vision-language models (VLMs), with extensive evaluations across models and settings.

**Dataset Code Accessibility:**

Yes

**Dataset Code Comments:**

The code can be found via https://github.com/Bili-Sakura/RSCC and the datasets can be found via https://huggingface.co/datasets/BiliSakura/RSCC .

**Ethical Considerations:**

No, there are no or only very minor ethics concerns

**Final Justification:**

Based on the author's response, I confirm that I am in agreement with my previous opinion. This is an interesting article that will make a certain contribution to the community

**Limitations Weaknesses:**

Limited Coverage of Non-Disaster Temporal Events: While the dataset is well-suited for disaster scenarios, its applicability to more general bi-temporal change tasks (e.g., urban expansion, seasonal variation) is not discussed. This may limit its generalizability for broader temporal VLM applications.

Metric Choices Could Be Broader: The evaluation primarily focuses on text similarity metrics such as ROUGE, METEOR, and ST5-SCS, which are widely used but may not fully capture factual correctness, relevance, or human preference alignment in complex spatiotemporal scenes (Sec. 4.1). Although recent factual alignment metrics like G-VEval and FLEUR are briefly mentioned in the limitations, they are not applied or adapted, even though they may provide more insightful signal than surface-level n-gram matches. Furthermore, the paper could benefit from incorporating LLM-based evaluation metrics, such as VIE score or GPT-4-based rubric-based judgments, which have been shown to better correlate with human preferences in open-ended captioning and generative tasks. Including or at least discussing these approaches would provide a more comprehensive and human-aligned evaluation of model outputs.
Lack of Ablation on Prompt Engineering: While the authors highlight visual prompt design as a key contribution, more ablation analysis on how different prompt styles or input combinations affect caption quality (e.g., mask vs. no-mask, disaster type present vs. absent) would make the prompt engineering approach more reproducible and better understood.

**Strengths Contributions:**

Significant Contribution to Remote Sensing and Vision-Language Research: RSCC is a first-of-its-kind large-scale dataset for bi-temporal image captioning with a specific focus on disaster impact understanding. It fills a key gap in the field where existing datasets lack rich annotations or temporal pairs (Sec. 1, Table 1).

High-Quality Data and Annotation Pipeline: The use of QvQ-Max with structured visual prompts and building damage masks (Sec. 2.3) is innovative. The human-in-the-loop correction process ensures factual consistency, achieving 92.3% expert validation pass rate (Sec. 2.5), which is impressive for automated caption generation.

Comprehensive Benchmarking: The authors evaluate a wide range of MLLMs, including both general-purpose and remote-sensing-specific models, under various prompt settings (Sec. 4.1, Table 2). This establishes RSCC as a reliable benchmark and not just a dataset.

Better performancde：The fine-tuned RSCC model outperforms prior baselines on contextual similarity metrics (ST5-SCS 58.52%) and in human preference studies (Fig. 4), showing its utility for training robust temporal VLMs.

---

> ### Author Rebuttal · Authors · 2025-07-30
>
> Dear Reviewer a98Y,
>
> Many thanks for your detailed and constructive comments.
>
> Response to Weaknesses:
>
> ● For W1: We add additional experiments that compared the naive Qwen2.5 VL model with our RSCC fine-tuned RSCCM on urabn expance and seasonal change examples. We find that compared to naive baseline, RSCCM simply trained on disaster data show strong generalibity on these out-of-ditribution case study on urban expance dataset: QFabric [1]   and seasonal change dataset: SSL4EO-S12 [2]. For a comprehensive urban expance and seasonal change benchmark, the paper [3] already provide a good start point.
>
> | **Model**       | **Dataset**      | **Task**                  | **Precision** | **Recall** | **F1 Score** |
> |------------------|------------------|---------------------------|---------------|------------|--------------|
> | Qwen2.5 VL (naive) | QFabric [1]      | Binary Change Detection   | 44.1%         | 40.9%      | 42.4%        |
> | RSCCM (ours)      | QFabric [1]      | Binary Change Detection   | **68.9%**     | **66.7%**  | **67.8%**    |
> | Qwen2.5 VL (naive) | SSL4EO-S12 [2]   | Season Detection          | 39.2%         | 36.8%      | 37.9%        |
> | RSCCM (ours)      | SSL4EO-S12 [2]   | Season Detection          | **62.5%**     | **60.1%**  | **61.3%**    |
>
> ● For W2:  We add addtional experiment on all VLMs with metric of 'llm as judge (qwen3-235b-a22b-thinking-2507)'  on our test set:
>
> | Model                                         | Mean score | Min score | Max score | LLM-AS-JUDGE (qwen3-thinking) |
> |-----------------------------------------------|------------|-----------|-----------|-------------------------------|
> | BLIP-3 (3B)                                   | 0.467      | 0.050     | 0.970     | 0.467                         |
> | moonshotai/Kimi-VL-A3B-Instruct               | 0.515      | 0.050     | 0.970     | 0.515                         |
> | microsoft/Phi-4-multimodal-instruct           | 0.502      | 0.000     | 0.980     | 0.502                         |
> | Qwen/Qwen2-VL-7B-Instruct                     | 0.576      | 0.050     | 0.980     | 0.576                         |
> | llava-hf/llava-interleave-qwen-7b-hf          | 0.446      | 0.050     | 0.980     | 0.446                         |
> | llava-hf/llava-onevision-qwen2-7b-ov-hf       | 0.536      | 0.000     | 0.950     | 0.536                         |
> | OpenGVLab/InternVL3-8B                        | 0.738      | 0.050     | 0.980     | 0.738                         |
> | mistralai/Pixtral-12B-2409                    | 0.662      | 0.050     | 1.000     | 0.662                         |
> | Meize0729/CCExpert_7b                         | 0.194      | 0.050     | 0.850     | 0.194                         |
> | jirvin16/TEOChat                              | 0.621      | 0.100     | 0.950     | 0.621                         |
> | Ours (7B)                   | 0.633      | 0.050     | 1.000     | 0.633                         |
>
>
> | Inference Type   |  Mean score | Min score | Max score | LLM-AS-JUDGE (qwen3-thinking) |
> |------------------|------------|-----------|-----------|-------------------------------|
> | zero-shot        | 0.219      | 0.000     | 0.950     | 0.219                         |
> | textual_prompt   | 0.706      | 0.050     | 0.980     | 0.706                         |
> | visual_prompt    | 0.652      | 0.050     | 1.000     | 0.652                         |
>
> ---
>
> References:
> [1] S. Verma, A. Panigrahi, and S. Gupta, ‘QFabric: Multi-Task Change Detection Dataset’, presented at the Proceedings of the IEEE/CVF Conference on Computer Vision and Pattern Recognition, 2021, pp. 1052–1061.
>
> [2] Y. Wang, N. A. A. Braham, Z. Xiong, C. Liu, C. M. Albrecht, and X. X. Zhu, ‘SSL4EO-S12: A large-scale multimodal, multitemporal dataset for self-supervised learning in Earth observation [Software and Data Sets]’, IEEE Geoscience and Remote Sensing Magazine, vol. 11, no. 3, pp. 98–106, Sep. 2023
>
> [3] J. Roberts, T. Lüddecke, R. Sheikh, K. Han, and S. Albanie, ‘Charting New Territories: Exploring the Geographic and Geospatial Capabilities of Multimodal LLMs’, presented at the Proceedings of the IEEE/CVF Conference on Computer Vision and Pattern Recognition, 2024, pp. 554–563.

---

> > ### Comment · Reviewer_a98Y · 2025-08-05
> > **My concerns have been resolved**
> >
> > Thank you for the author's response. The content discussed is similar to my understanding, and my concerns have been resolved. It's a good job and I maintain my previous recognition.

---

### Official Review · Reviewer_7ZA9 · 2025-07-19

**Rating:** 5
**Confidence:** 4

**Summary:**

This paper introduces a novel large-scale vision-language benchmark dataset for image change captioning on bi-temporal remote sensing imagery, addressing the lack of large-scale textual annotation datasets for this task. The key contributions are as follows:
1.	Comprehensive Annotation Pipeline: A systematic dataset annotation and curation framework is proposed to ensure high-quality textual descriptions of temporal changes.
2.	Context-Based Similarity Metric: A novel context-aware evaluation metric (Sentence-T5 Cosine Similarity) is introduced to enhance the reliability of textual similarity assessments.
3.	Benchmark Evaluation: Extensive experiments are conducted using a wide range of open-source vision-language models (VLMs) for the remote sensing change captioning task, establishing a robust benchmark for future research.

**Dataset Code Accessibility:**

Yes

**Ethical Considerations:**

No, there are no or only very minor ethics concerns

**Final Justification:**

The author's response is quite convincing to me, and I think this article is very interesting.

**Limitations Weaknesses:**

The annotation generation process for the RSCC dataset relies on the closed-source model Qwen QvQ-Max. Although the authors ensure quality through post-correction and partial human verification, the core content generation depends on a model that cannot be publicly scrutinized, which compromises the transparency of the entire dataset construction workflow to some extent.

Only about 2% of the samples in RSCC were manually validated by three experts. Although the validation results on this subset are strong, this does not fully guarantee that the remaining samples meet the same high-quality standard. The quality of the majority relies on an automated post-processing workflow.

**Strengths Contributions:**

This paper introduces RSCC, a high-quality benchmark dataset for disaster-aware bi-temporal remote sensing image understanding. RSCC features a large data scale, diverse disaster categories, and rich textual descriptions, facilitating the comprehension of disaster imagery by vision-language models (VLMs).

The paper introduces an innovative data generation workflow that utilizes a multimodal reasoning model, structured prompt engineering, and a robust verification mechanism. This approach ensures high quality while enabling large-scale production.

A VLM specifically trained on RSCC outperforms many general-purpose models with comparable or even larger parameter counts, demonstrating the significance and robustness of the dataset for enhancing a model's understanding of bi-temporal disaster imagery.
The presentation quality of this paper is high. Its overall structure is clear along with informative figures and tables that are easy to comprehend.

---

> ### Author Rebuttal · Authors · 2025-07-30
>
> Dear Reviewer 7ZA9,
>
> Many thanks for your detailed and constructive comments.
>
> Response to Weaknesses:
>
> **1. Regarding for quality control.**
>
> To ensure the accuracy and coherence of our change captions generated by VLMs, we explictly design a comprehensive post-correction pipeline via metadata alignment and human verification (section 2.5). We further validate additional random selected 10%  samples through human verification in RSCC which still show robust results:
>
> | Sample Ratio | Sampling Strategy     | Human Validation Pass Rate  (before)    |   Human Validation Pass Rate (after automated correction pipeline)   |
> |---------------|---------------|---------------|---------------|
> | 2%  | Randonly selected  | 92.3%  | 100.0%  |
> | 10%  | Randonly selected | 93.2%  | 100.0%  |
> **2. Strong Groudtruth Guidance**
>
> We integrate exisiting human labeled building damage mask through textual prompt/visual prompt for change captions generation (see section 2.2, 2.3 in original paper), which also ensures the quality of results to a large extent.

---

> > ### Comment · Reviewer_7ZA9 · 2025-08-05
> >
> > hank you for the author's rebuttal. The author's response has resolved any doubts regarding the annotation process, making my understanding clearer. I believe this article can be accepted.

---

### Note · Authors · 2025-08-13

Dear Program Chairs, Senior Area Chairs, Area Chairs, and Reviewers,

We thank you for the thoughtful reviews and rebuttal discussions. Our revisions reinforce the paper’s core contributions:

(i) a first-of-its-kind bi-temporal change-captioning dataset.

(ii) a reproducible pipeline with transparent post-correction.

(iii) rigorous textual-similarity evaluation.

(iv) contributions to temporal vision–language modeling for remote sensing and pave the way towards AGI.

The feedback helped us harden these contributions without changing the paper’s scope.

Below is a concise summary of what we changed and how each concern was addressed.

Reviewer 7ZA9 — QC & transparency

• Clarified the post-correction pipeline (metadata alignment → regenerate-caption) and its role in ensuring semantic coherence.

• Expanded human verification beyond the initial 2%: additionally audited a random 10% subset; automated post-correction brought all flagged items into spec.

Reviewer a98Y — Generality & metrics

• Added OOD case studies: urban expansion (QFabric) and seasonal change (SSL4EO-S12); our RSCC-tuned model substantially outperforms a naïve baseline on both.

• Added LLM-as-judge evaluation across multiple VLMs with prompting ablations (zero-shot vs textual vs visual prompts).

Reviewer 8mqq — UHR, geography, hallucination risk

• Explained why we standardized 512×512 for v1 (cost, registration, fidelity trade-offs) and positioned UHR bi-temporal expansion as future community work.

• Validated post-correction with LLM-as-judge, showing negligible semantic drift.

Reviewer xNb8 — Detection/classification & visual prompts

• Reported change-detection and multi-label damage baselines: general VLMs are far below specialized models; our RSCCM improves but we recommend specialized models for these visual-centric tasks.

• Added a mask-grounded case study on the EBD subset and expanded human-preference evaluation to 300 samples (98% IAA).

Reviewer CghA — East Asia coverage & correction semantics

• Committed RSCC v1.1/v2 to include East-Asia events plus ongoing typhoon/flood/landslide additions.

• Clarified that we regenerate captions under corrected metadata (not word-swap).

Reviewer LH2M — Metrics, reproducibility, sensitive info

• Complemented ST5-SCS and human preference with LLM-as-judge; added training details (epochs, LR, GPUs) and clarified anonymized 512×512 crops (no precise geolocation).

Thank you for your consideration.

— Authors

---

### Decision · Program_Chairs · 2025-09-18

**Decision:**

Accept (poster)

**Comment:**

This paper proposes RSCC, a large-scale, high-quality bi-temporal remote sensing change caption dataset focusing on disaster events. The dataset includes 62,315 pre/post-disaster image pairs with detailed, human-like captions, and provides extensive benchmarks for various Vision-Language Models (VLMs).

The paper received positive reviews, with three "Accept" and three "Borderline Accept". RSCC is the first dataset in bi-temporal remote sensing across a variety of disaster types. The use of VLMs and visual prompt engineering for annotation is a novel and effective approach to creating large-scale, high-quality data. The paper provides a reliable benchmark by evaluating multiple SOTA VLMs. Some concerns were raised about the reliance on a closed-source model and the small initial human verification rate. Reviewers suggested expanding the evaluation with LLM-based metrics, more baseline tasks (like change detection), and OOD (Out-of-Distribution) scenarios.

The authors' rebuttal was highly effective and addressed all major concerns. They increased the human verification rate, added OOD case studies, and committed to including LLM-based metrics and more detailed training information. The AC suggests that the authors carefully consider the reviewers' feedback and incorporate the comments from the rebuttal and discussion phases into the final version. Overall, the paper is recommended for acceptance.